# ExoSTING, an extracellular vesicle loaded with STING agonists, promotes tumor immune surveillance

Su Chul Jang [1], Kyriakos D. Economides[1], Raymond J. Moniz[1], Chang Ling Sia[1], Nuruddeen Lewis[1], Christine McCoy[1], Tong Zi[1], Kelvin Zhang[1], Rane A. Harrison[1], Joanne Lim[1], Joyoti Dey[2], Marc Grenley[2], Katherine Kirwin[1], Nikki L. Ross[1], Raymond Bourdeau [1], Agata Villiger-Oberbek[1], Scott Estes[1], Ke Xu[1], Jorge Sanchez-Salazar[1], Kevin Dooley[1], William K. Dahlberg[1], Douglas E. Williams[1] & Sriram Sathyanarayanan [1]✉

Cyclic dinucleotide (CDN) agonists of the STimulator of InterferoN Genes (STING) pathway have shown immune activation and tumor clearance in pre-clinical models. However, CDNs administered intratumorally also promote STING activation leading to direct cytotoxicity of many cell types in the tumor microenvironment (TME), systemic inflammation due to rapid tumor extravasation of the CDN, and immune ablation in the TME. These result in a failure to establish immunological memory. ExoSTING, an engineered extracellular vesicle (EV) exogenously loaded with CDN, enhances the potency of CDN and preferentially activates antigen presenting cells in the TME. Following intratumoral injection, exoSTING was retained within the tumor, enhanced local Th1 responses and recruitment of CD8+ T cells, and generated systemic anti-tumor immunity to the tumor. ExoSTING at therapeutically active doses did not induce systemic inflammatory cytokines, resulting in an enhanced therapeutic window. ExoSTING is a novel, differentiated therapeutic candidate that leverages the natural biology of EVs to enhance the activity of CDNs.

[1] Codiak BioSciences Inc., Cambridge, MA, USA. [2] Presage Biosciences, Seattle, WA, USA. ✉email: sriram.sathy@codiakbio.com

Cancer immunotherapy with checkpoint inhibitors can induce long-lasting objective tumor responses in patients with metastatic cancers of a wide range of histologies[1,2]. However ~85% of patients fail to benefit from these therapies, with lack of T cell infiltration and immune recognition identified as key mechanisms responsible for lack of efficacy[1]. Strategies to improve immunotherapy outcomes involve methods to stimulate innate immune response pathways. Promising preliminary clinical results have been seen with toll-like receptor (TLR) pathway agonists and oncolytic viruses[3,4], suggesting that this strategy has merit. Another well-studied and critical pathway of the innate immune response is the STimulator of InterferoN Genes (STING) pathway. Activation of the STING pathway occurs by DNA recognition by cyclic guanosine monophosphate synthetase (cGAS) resulting in the generation of the cyclic dinucleotide (CDN) STING ligand 2′3′ cGAMP and is required for the generation of tumor-specific T cell responses, representing one mechanism of immune surveillance in the TME[5]. This effect is mediated by induction of cytokines, including type I interferons and chemokines[6]. Based on the critical role of the STING pathway in anti-tumor immunity, multiple synthetic STING activators are being explored for cancer therapy[5,7].

STING activation with various CDN and non-CDN agonists has demonstrated potent tumor regression in several preclinical tumor models[8,9]. Preclinical and clinical studies using intratumorally (IT) injected CDNs consistently demonstrate bell-shaped dose response curves, wherein systemic immunity is established at lower doses but is lost at higher doses[10,11]. Preclinical studies using CDNs have demonstrated dose-dependent elimination of tumor-infiltrating effector T cells and a subsequent inability to establish systemic anti-tumor immunity[12]. The low membrane permeability, short residence time, and lack of specific uptake into antigen presenting cells (APCs) following IT administration necessitate high CDN doses to achieve the requisite exposure in the TME, which prevent the establishment of tumor memory T cell responses. This delicate balance between effective immune stimulation and immune ablation creates a significant challenge to defining optimal therapeutic doses of CDNs.

Extracellular vesicles (EVs) have been identified as natural mediators of signaling between cancer cells and tumor resident APCs[13]. In particular, EVs have been shown to contain tumor-derived dsDNA and shuttle this cargo to tumor resident dendritic cells (DCs)[14,15], which activates a type I IFN response via the STING pathway[14]. DNA-containing EVs derived from cancer cells treated with topotecan have also been shown to activate the STING pathway and reinforce anti-tumor immunity[15]. These observations regarding the natural biology of EVs in the TME and their potent activation of the STING pathway prompted us to use our molecular engineering platform[16] to design a therapeutic EV candidate called exoSTING. ExoSTING consists of an engineered EV loaded with a potent CDN STING agonist. ExoSTING demonstrates greater than 100-fold increased potency in in-vivo tumor models and has increased tumor retention and lower levels of systemic inflammatory cytokine production as compared to free CDN, expanding the therapeutic index. Furthermore, we have demonstrated enhanced Th1 polarization of T cells with exoSTING and preservation, expansion, and establishment of systemic antigen-specific T cell-mediated immune responses across a wide dose range without evidence of immune ablation as seen with free CDNs. ExoSTING represents a novel strategy to harness and improve upon natural immune surveillance in the TME and may overcome many of the observed limitations with dose selection of free CDNs, which are currently in the early stages of clinical development[10,11].

## Results

### EV-mediated delivery enhances potency and anti-tumor activity of CDN STING agonists.

Engineered EVs have emerged as an efficient delivery system, leading to higher drug accumulation in target cells and improved potency[17]. We hypothesized that EV-mediated delivery of CDN may significantly increase potency and preferential activation of APCs in the TME. To this end, we developed a methodology to purify a population of EVs stringently and reproducibly from large volumes of cell culture supernatant, similar to those recently reported by Jeppesen et al.[18]. EVs were purified by ultracentrifugation and discontinuous density gradient from cultured medium of wild type HEK293 cells or cells overexpressing Prostaglandin F2 receptor negative regulator (PTGFRN) (Supplementary Fig. 1a). EVs in the lower density F1 fraction ranged in size approximately 50–200 nm, as measured by electron microscopy and nanoparticle tracking analysis (Supplementary Fig. 1b, c). It should be noted that the higher density F2, F3, and F4 fractions contain mostly proteinaceous, non-vesicular material predominantly of ECM proteins and histones[16]. Purified EVs contained high levels of PTGFRN and other canonical EV markers including CD9, CD63, CD81, Alix, and TSG101, but devoid of the endoplasmic reticulum protein such as Calnexin (Supplementary Fig. 1d). HEK293 lineage was selected for EV production specifically for its well-established documentation of subject safety associated with the cell line[19], and the capacity to conduct cell culture in chemically defined medium devoid of contaminating EVs from animal serum. Furthermore, HEK293 cell derived EVs that are highly purified by density gradient method showed no endogenous immune modulatory effects. These native EVs did not induce stimulation of T cells (CD4 or CD8), B cells as measured by CD69 activation, or monocytes as measured by CD80 activation, even at very high EV to cell ratios (Supplementary Fig. 1e). The oncoprotein E1A was present at very low levels in the purified EV preparation (Supplementary Fig. 1f). Engineered HEK293 derived EVs overexpressing PTGFRN were utilized for further studies due to reproducible, scalable manufacturing, and quality attributes required to support clinical translation of exoSTING[20].

Two different CDNs, CDN1 (ML RR-S2, a 2′-3′ CDN) or CDN2 (cAIM(PS)2 Difluor, a 3′-3′ CDN) were loaded into EVs and excess CDN was removed (Supplementary Fig. 2a). CDN content in the EV was quantified using mass spectrometry. The number of CDNs per EV (1189 ± 382 or 988 ± 339 for CDN1 or CDN2, respectively) and size distribution of loaded EVs were similar between CDN1 and CDN2 (Supplementary Fig. 2b, c). It should be noted that the loaded CDNs may be associated with inside and/or outside of EV. We characterized the potency of exoSTING in an in vitro PBMC assay. Unloaded EVs did not induce any IFN-β (Supplementary Fig. 3a). Free CDN1 or CDN2 resulted in IFN-β production with an $EC_{50}$ ~ 9.7 μM compared to 0.1 μM for CDN1 or CDN2-loaded EVs (Fig. 1a and Supplementary Fig. 3b). ExoSTING was ~100-fold more potent (Fig. 1a; lower $EC_{50}$) than free CDN, regardless of the specific CDN used. Similar improvements in potency were observed with EVs loaded with CDN1 across multiple donors ($n = 12$; Fig. 1b). Notably, the potency enhancement was maintained with exoSTING that was preserved at −80 °C up to 1 year (Supplementary Fig. 3c). A similar potency increase was observed with EVs derived from mesenchymal stem cells loaded with CDN2 (Supplementary Fig. 3d). Addition of impurities from the F4 fraction profoundly inhibited the activity of exoSTING (Supplementary Fig. 3e) highlighting the importance of EV purity required for potency.

Next, we determined if loading EVs with CDNs is required for enhancement of potency. We compared the activity of co-administered EVs and free CDN with exoSTING in vitro. The results showed that $EC_{50}$ values for IFN-β production were

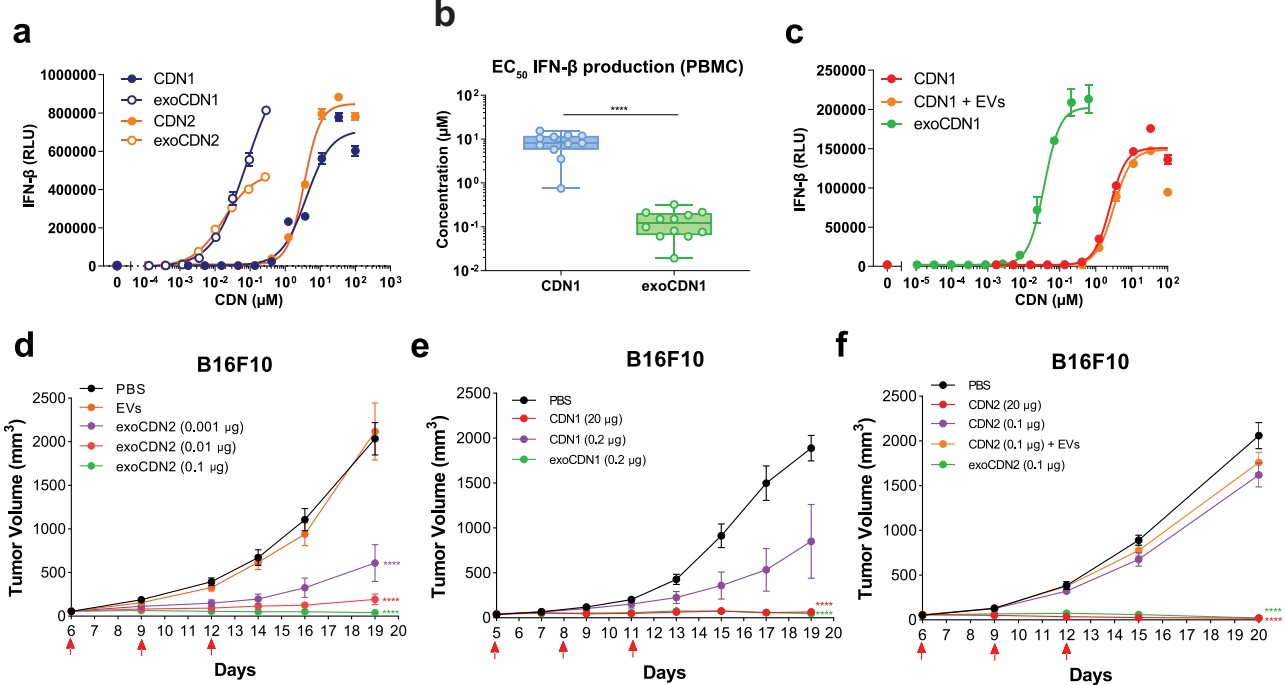

**Fig. 1 exoSTING enhances the potency of a CDN in in vitro and in vivo. a** Representative dose-response curves of IFN-β production in human PMBC supernatant after treating with two different CDN-loaded EVs, exoCDN1 and exoCDN2, compared with free CDNs ($n = 2$ biological replicates per donor). Biological replicates (three healthy donors) are presented in Supplementary Fig. 2b. **b** EC$_{50}$ value of IFN-β production in human PBMC after treating with free CDN1 and exoCDN1 ($n = 12$ healthy donors). **c** Representative dose-response curves ($n = 3$ healthy donors) of IFN-β production in PBMC after treating with free CDN1, free CDN1 with EVs, and exoCDN1 ($n = 2$ biological replicates per donor). ****$P < 0.0001$ by unpaired $t$-test. **d–f** C57BL/6 mice were implanted subcutaneously with $1 \times 10^6$ B16F10 cell on right flank of mice. Additional implantation of tumors is specified in each legend. When tumor volumes reached 50–100 mm³, testing agents were injected intratumorally three times with 3 days interval. Red arrows in the graph indicate IT injection days. Tumor growth were measured over time ($n = 5$ animals per group). CDN1 MR SS-2 CDA, CDN2 cAIM(PS)2 Difluor (Rp/Sp), RLU relative luminescent unit. Data are presented as means ± s.e.m from replicate samples as indicated. ****$P < 0.0001$ by two-way ANOVA with Tukey's multiple comparison test for tumor growth studies.

similar between free CDN (2.5 μM) and co-administered EVs with free CDN (3.1 μM), whereas exoSTING showed ~100-fold improvement in potency (0.03 μM) (Fig. 1c). In addition, competition assay with unloaded EVs inhibited exoSTING mediated IFN-β production in a dose-dependent manner and reached to background level at >50× concentration of the loaded EVs (Supplementary Fig. 3f). Although we have not fully established the mechanism of EV-mediated CDN dependent STING activation, these data demonstrate that loading of CDN into EVs is required for the observed enhancement of potency. To assess immune cell subsets activated by free CDN or exoSTING, we evaluated immune cell activation markers by flow cytometry in an in vitro PBMC assay, with CD86 as an activation marker for DCs and monocytes. ExoSTING treatment resulted in activation of cDCs at 1000-fold lower doses of exoSTING as compared to the free CDN (EC$_{50}$ ~0.0001 μM vs. ~1.2 μM) and enhanced potency was also observed in monocytes (Supplementary Fig. 3g, h).

For evaluation of anti-tumor efficacy, we selected the subcutaneous B16F10 model as a "cold" tumor (Supplementary Fig. 4a, b) that is devoid of T cell infiltration and has been shown to be refractory to checkpoint inhibitor therapy[21,22]. We compared the efficacy of IT injections of exoSTING and free CDN to determine whether the observed in vitro potency enhancements would translate in vivo. When tumors reached 50–100 mm³, IT dosing was performed as indicated. Our results demonstrated the dose-dependent tumor growth inhibition with exoSTING at doses 200–300-fold lower than that required with free CDN1 or CDN2 (Fig. 1d, e). Similar enhanced potency of exoSTING was observed in multiple tumor models including

EG7.OVA and CT26.wt tumors (Supplementary Fig. 4c, d). No tumor growth inhibition was observed with unloaded EVs demonstrating that the CDN is required for in vivo anti-tumor activity (Fig. 1d and Supplementary Fig. 4e, f).

To evaluate if loading of CDNs into EVs is required for improved anti-tumor activity of exoSTING, we compared free CDN2 with a mixture of CDN2 and EVs co-administered in the B16F10 model. Neither free CDN2 (0.1 μg) or co-administered unloaded EVs mixed with free CDN2 (0.1 μg) resulted in measurable tumor growth inhibition (Fig. 1f). Both in vitro and in vivo studies demonstrate that loading of STING agonists into EVs enhanced the potency of CDNs.

**exoSTING results in systemic tumor growth control**. Using two B16F10 abscopal tumor models, we determine the effect of exoSTING on systemic anti-tumor immunity. First, a primary B16F10 tumor was inoculated subcutaneously in the flank followed by an intravenous injection of B16F10 cells, which induced distal lung lesions. ExoSTING was administered into the subcutaneous tumors and both the injected primary tumor and the lung lesions were monitored. Consistent with published results[12], high doses of free CDN (20 μg) were required to inhibit the primary tumor in the flank, while tumor growth inhibition was observed at >100-fold lower doses of exoSTING (0.12–0.012 μg) (Fig. 2a). Histological analysis of revealed the presence of multiple tumor lesions in the control group (Fig. 2b, left). ExoSTING treatment resulted in pathologically relevant complete remission (CR) in the lung (Fig. 2b, right). Mice treated with high doses of free CDN showed many viable tumor

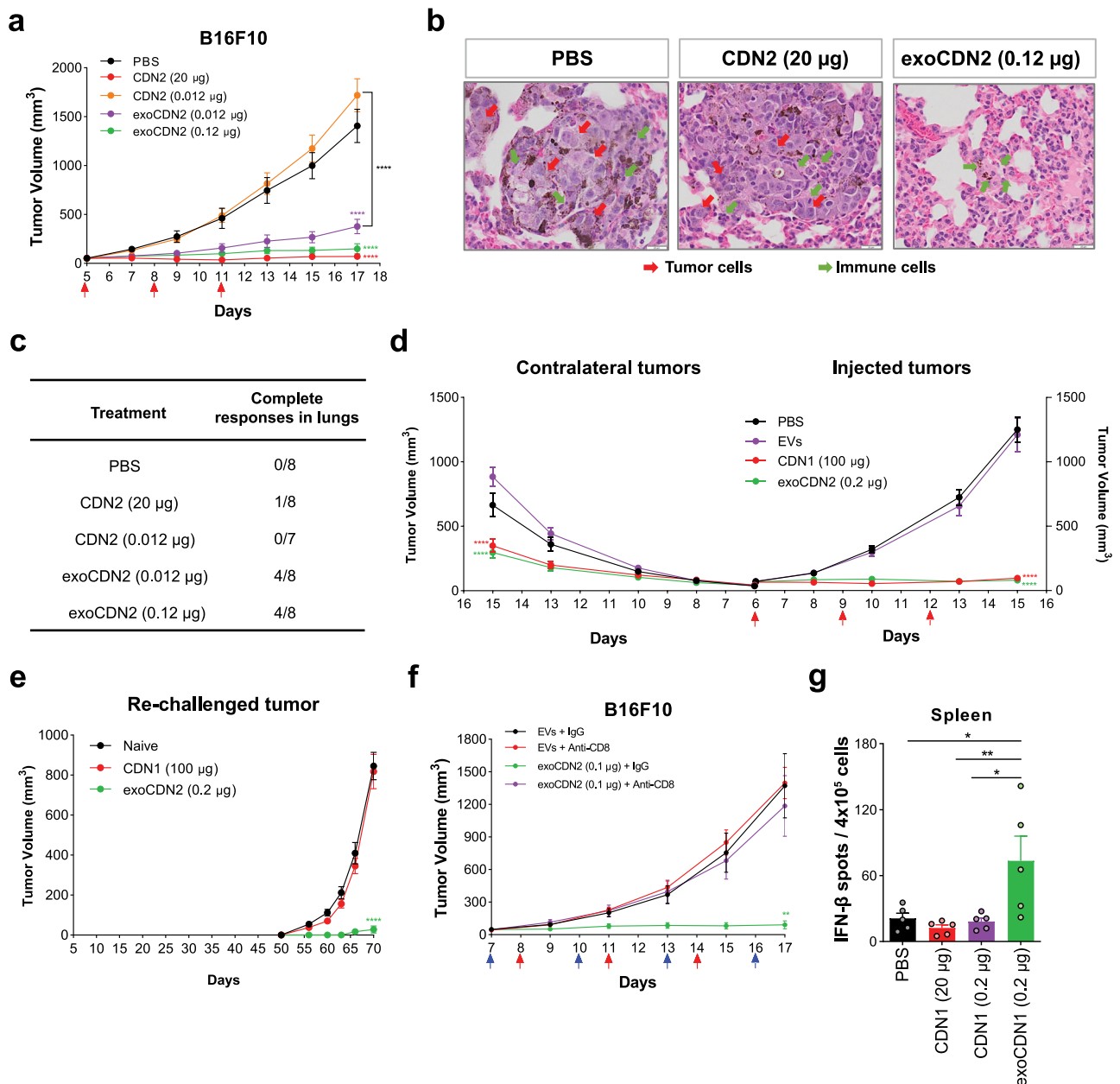

**Fig. 2 exoSTING elicits strong anti-tumor responses with systemic tumor-specific immune activation in a B16F10 tumor model. a–c** B16F10 tumor cells (1 × 10^6 cells) were implanted subcutaneously in right flank of C57BL/6 mice. At Day 4, B16F10 tumor cells (1 × 10^5 cells) were injected intravenously to induce lung metastasis. PBS, free CDN2 (20 or 0.012 μg) and exoCDN2 (0.012 or 0.12 μg) were injected into subcutaneous tumors (*n* = 8 animals per group). Tumor growth curves of subcutaneous tumors (**a**), representative H&E stained images from PBS, CDN2 (20 μg), and exoCDN2 (0.12 μg) (**b**), and number of complete responders by histopathological analysis (**c**). Complete remission defined by pathologist from H&E staining of whole lungs. **d** B16F10 tumor cells were implanted subcutaneously in right (1 × 10^6 cells) and left (5 × 10^5 cells) flanks of mice (*n* = 10 animals per group). PBS, unloaded EVs, CDN1 (100 μg), exoCDN2 (0.2 μg) were injected intratumorally into right flank tumors and both injected and non-injected contralateral tumor growth was measured. **e** Subcutaneous B16F10 tumors were treated with PBS (*n* = 5 animals), unloaded EVs (*n* = 5 animals), CDN1 (100 μg) (*n* = 10 animals), and exoCDN2 (0.2 μg) (*n* = 15 animals). B16F10 cells (1 × 10^6 cells) were implanted to mice that had CR and naïve mice (*n* = 5 animals) on day 50 to the left flank. Re-challenged tumor growth was measured. **f** Tumor growth inhibition after CD8 T cell depletion. B16F10 tumor bearing mice received the IgG (10 mg/kg) or anti-CD8 antibody (10 mg/kg) intraperitoneally, one day before IT injection (*n* = 5 animals per group). Blue arrows indicate intraperitoneal injection days. **g** Tumor specific IFN-γ response after PBS, CDN1 (0.2 or 20 μg), and exoCDN1 (0.2 μg) treatment to B16F10 tumor (*n* = 5 animals per group). Data are presented as means ± s.e.m from replicate samples as indicated. *P < 0.05; **P < 0.01; ***P < 0.001; ****P < 0.0001 by one-way ANOVA for **g** and two-way ANOVA with Tukey's multiple comparison test for tumor growth studies.

cells with little T cell infiltration in the lung (Fig. 2b, middle). It should be noted that this dose of free CDN (20 μg) completely abrogated primary tumor growth. Importantly, 4 out of 8 mice (50%) in the exoSTING group demonstrated a histological CR with no evidence of lung tumors at dose levels evaluated

(Fig. 2c). In contrast, only one of eight mice (12%) in the high dose-free CDN group showed CR. These results highlight the greater potency of exoSTING at the injected tumor sites and enhanced capacity to induce a systemic immune response against distant tumors.

We next evaluated exoSTING in a second abscopal model by monitoring tumor growth after implanting B16F10 tumors on both flanks and injecting only the primary tumor. ExoSTING (0.2 µg) injection showed robust inhibition of growth not only in the injected primary tumor but also in the non-injected contralateral tumors, whereas unloaded EVs did not inhibit the growth (Fig. 2d). As 20 µg of CDN was ineffective in controlling the non-injected tumor growth in this model (Supplementary Fig. 5a)[12], in agreement with the observed lack of efficacy in the abscopal lung model, we used a very high dose of free CDN (100 µg) for effective control of both the injected and non-injected tumors (Fig. 2d). This control of the non-injected tumors could be either due to the establishment of systemic T cell immunity, or due to the systemic leakage of the free CDN and exposure to the distal non-injected tumor. Previous studies with free CDN1 had demonstrated that, at these very high doses, accumulation of drug in the secondary non-injected tumor and immune ablation can occur[12]. To distinguish between these two different mechanisms, we evaluated immunological memory response using a B16F10 re-challenge model. B16F10 tumor bearing mice were treated with doses of either exoSTING (0.2 µg) or free CDN (100 µg). Both treatments resulted in growth control of the primary injected tumor. Persistent tumor growth control was monitored over a period of 50 days. Complete remission was observed in one-third of the mice treated with exoSTING and 80% treated with free CDN (100 µg) (Supplementary Fig. 5b). Mice that showed complete remission were subsequently re-challenged on the opposite flank with a second inoculation of B16F10 tumor cells on day 50. No tumor growth was observed upon tumor challenge with B16F10 cells in the exoSTING treated group up to day 70 (Fig. 2e), however, tumor growth was observed in all of the mice previously treated with high dose of free CDN demonstrating a lack of immunological memory (Fig. 2e). These data suggest that local and systemic anti-tumor activity of free CDN and exoSTING may be mediated by distinctly different mechanisms.

**exoSTING generates tumor antigen-specific CD8 T cell-dependent systemic anti-tumor immunity**. To determine the requirement of different innate and adaptive immune cell types in exoSTING-mediated anti-tumor activity, we depleted CD8+ T cells, NK cells, and macrophages by treating mice with depleting antibodies. CD8+ T cells play a central role in mediating anti-tumor immunity by exoSTING as demonstrated by the abrogation of anti-tumor activity following selective antibody-mediated depletion of these cells (Fig. 2f). Tumor growth inhibition with exoSTING was not affected by NK cell depletion (Supplementary Fig. 5c). Depletion of tumor-associated macrophages resulted decreased in efficacy and only 50% tumor growth inhibition (Supplementary Fig. 5d). These results demonstrate the essential role of CD8+ T cells in the anti-tumor activity of exoSTING.

We next evaluated the capacity of exoSTING to induce systemic, antigen specific T cell responses in the B16F10 tumor model. Several tumor-associated CD8-dependent antigens have been identified as dominant antigens from B16F10 tumor cells[23]. We used an equimolar mixture of the epitope peptides (Trp2, GP100, and Tyr) to stimulate tumor antigen-specific T cell responses by assessing IFN-γ production in splenocytes following two IT doses of exoSTING or free CDN1. At day 4, 24 h after the second dose, exoSTING (0.2 µg) demonstrated a 3-fold to 4-fold increase in the number of IFN-γ positive spots as compared to the PBS control (Fig. 2g). The equivalent dose of free CDN1 (0.2 µg) or the efficacious dose (20 µg) failed to induce IFN-γ responses. These data demonstrate that exoSTING treatment results in robust expansion of tumor antigen-specific T cells in contrast to free CDN.

**exoSTING is retained in tumors, enhances local IFN-γ induction, and limits systemic inflammation**. A major limitation of IT administration of free CDN is its rapid dissemination into the systemic circulation and systemic inflammation[12]. The amount of CDN in B16F10 tumors and blood was measured over time after a single IT dose. The pharmacokinetic parameters from the study are summarized in Supplementary Table 1. Although both exoSTING and dose matched free CDN2 (0.3 µg) had a similar maximum concentration ($C_{max}$), the area under the concentration-time curve (AUC) of exoSTING is approximately 10-fold higher, the half-life ($T_{1/2}$) is longer (4.7-fold), and clearance is slower (~10-fold), demonstrating increased tumor retention of exoSTING as compared with an equivalent dose of free CDN2 (Fig. 3a). As previous studies have demonstrated that high doses of free CDN (20–100 µg) were required for efficacy[12], we also evaluated a 30 µg dose of free CDN2, which rapidly cleared from the injected tumor and was detectable in the plasma by 30-min. In contrast, exoSTING at the therapeutically active and maximum feasible dose (0.3 µg) was just above the lower limit of quantitation (LLOQ) of the assay at 5 min (9.4 ng/mL) and not detectable at 30 minutes (Fig. 3b). These data confirm prolonged tumor exposure and limited systemic exposure observed at the maximal feasible dose with exoSTING.

To measure the pharmacodynamic impact of prolonged CDN retention and STING activation in exoSTING injected tumors, we analyzed intra-tumoral mRNA levels of IFN-β, and Th1 regulated chemokines CXCL9 and CXCL10 four hours post-injection of exoSTING or free CDN into B16F10 tumors. ExoSTING induced four-fold higher levels of IFN-β in the TME compared to equivalent doses of free CDN. To achieve comparable levels of intra-tumoral IFN-β, free CDN treatment required a 100-fold higher dose than exoSTING (Fig. 3c). ExoSTING also induced substantially higher levels of CXCL9 (5-fold vs free CDN) and CXCL10 (3-fold vs free CDN) mRNA than a comparable amount of free CDN (Fig. 3d, e). Doses of free CDN, even when dosed 100-fold higher than exoSTING, failed to induce comparable mRNA levels of CXCL9 and CXCL10. Importantly, unloaded EVs did not change the IFN-β, CXCL9, and CXCL10 mRNA expression (Supplementary Fig. 6a).

We compared inflammatory cytokine levels in the blood following IT administration of exoSTING or free CDN and found that while 0.2 µg of exoSTING or free CDN failed to induce systemic cytokines, the efficacious dose of free CDN (20 µg) resulted in pronounced induction of inflammatory cytokines (IFN-β, TNF-α, and IL-6) (Fig. 3f–h). In addition, cytokine upregulation was observed in the draining lymph nodes and spleen by high dose free CDN (20 µg) but not by low dose free CDN or exoSTING (Supplementary Fig. 6b, c). The lack of systemic inflammatory cytokine induction following an efficacious dose of exoSTING may reduce adverse events noted in early clinical testing with free CDNs while maximizing exposure in the TME[14,15].

**exoSTING preserves immune cell viability and enhances DC activation and T cell recruitment**. To distinguish the mechanisms of action, we histologically evaluated the immune cell infiltrate following IT dosing of therapeutically active doses of free CDN (20 µg) or exoSTING (0.1 µg) in the B16F10 model. Free CDN2 treatment showed evidence of tissue damage in the skin and tumor cell death at both 4 and 24 h after two doses, while there was limited tissue damage observed in exoSTING treated tissues (Fig. 4a, top row). At 4 h after the second dose,

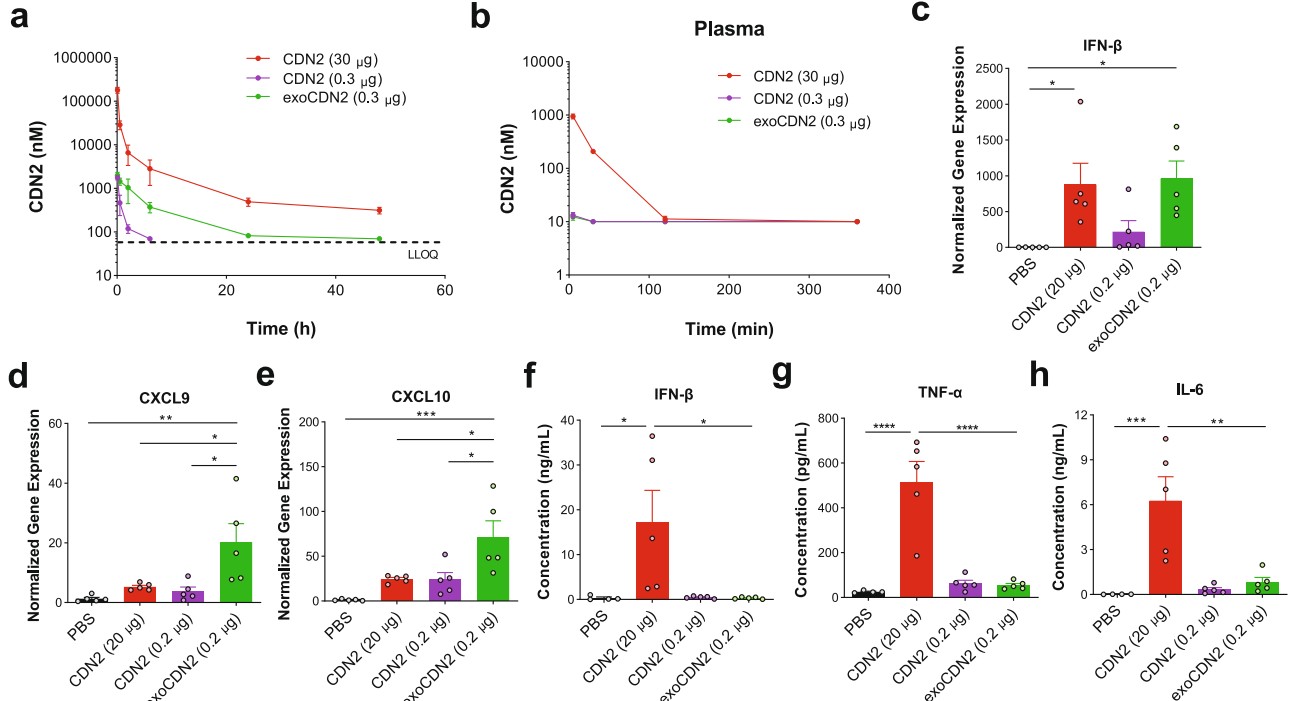

**Fig. 3 Intra-tumoral administration of exoSTING enhances pharmacokinetics of a CDN and immunostimulatory activity in tumor microenvironment.**
**a–h** C57BL/6 mice were implanted subcutaneously with $1 \times 10^6$ B16F10 cells. Concentration of CDN2 in tumors (**a**) and plasma (**b**) was measured by LC-MS/MS at 5 and 30 min, 2, 6, 24, and 48 h after single IT injection ($n = 3$ animals per group at each time point, $n = 6$ animals for CDN2 (30 μg) and exoCDN2 (0.2 μg) at 5 and 30 min, 2 and 6 h). **c–e** Four hours after IT injection, RNAs were purified from tumors ($n = 5$ animals per group) and serum was collected. Relative expression of IFN-β (**c**), CXCL9 (**d**), and CXCL10 (**e**) genes was measured by RT-qPCR, normalized against the housekeeping gene RPS13. Serum levels of IFN-β (**f**), TNF-α (**g**), and IL-6 (**h**) were measured ($n = 5$ animals per group). Data are presented as means ± s.e.m from replicate samples as indicated. *, $P < 0.05$; **, $P < 0.01$; ***, $P < 0.001$; ****, $P < 0.0001$ by one-way ANOVA with Tukey's multiple comparison test.

exoSTING induced comparable levels of IFN-β as the first dose, but lower induction was observed with free CDN, suggesting that the tissue damage induced by high dose free CDN impairs IFN-β production and results in reduced immune cell infiltration (Fig. 4a, middle row; Fig. 4b). The observation of local tissue damage around the injection site and immune ablation with free CDN agonists is consistent with previous reports[12]. At 24 h post dose there was minimal IFN-β remaining (Fig. 4a, middle row; Fig. 4b) consistent with the tight regulation of IFN-β production following STING agonism. There were minimal levels of IFN-β following treatment with unloaded EVs (Fig. 4b). Most of IFN-β production correlated with F4/80 in the exoSTING-treated tumors, whereas IFN-β production was observed in both F4/80 positive and F4/80 negative cells in free CDN-treated tumors demonstrating the preferential activation of macrophages by exoSTING (Fig. 4c). We observed that tumors treated with 20 μg of free CDN had significantly lower levels of infiltrating T cells and F4/80+ APCs than tumors treated with exoSTING (Fig. 4a, bottom row), which had four-fold more infiltration of CD8+ T cells than observed in control tumors (Fig. 4d). This result highlights the increased recruitment of CD8 T cells to the tumor microenvironment by exoSTING.

When we assessed CD8+ T cell and XCR1+ DC activation by flow cytometry, we found that two IT doses of free CDN (20 μg) resulted in ablation of T cells (Fig. 4e) as well as reduction of both F4/80 positive macrophages and DCs (Supplementary Fig. 7). In contrast, we observed an increase (>1.5 fold) in CD8+ T cells over control treatment (Fig. 4d). ExoSTING treatment resulted in significant activation of XCR1+ BATF3 lineage DCs as measured by CD86 expression, whereas free CDN did not activate these DCs (Fig. 4f). This DC population has been shown to be essential for establishing a systemic anti-tumor immune response[5]. The

enhanced activation of this DC subset seen with exoSTING is consistent with the observed superiority of anti-tumor activity of exoSTING in the B16F10 model (Figs 1 and 2). These data demonstrate the immune ablative effects of free CDN and highlight the improved immune stimulatory effects of exoSTING.

**Differential gene expression studies confirm potent immune stimulation by exoSTING.** To characterize the immune pathway activation, we evaluated the global gene expression changes in B16F10 tumors after 1 or 2 IT injections of efficacious doses of exoSTING (0.1 μg) and free CDN2 (20 μg) by RNA sequencing. Expression profiles after IT injection of unloaded EVs were not significantly changed demonstrating that EVs alone do not elicit significant global gene expression changes (Fig. 5a). Genes belonging to Th1 activation pathways were enriched only in exoSTING treated tumors (Fig. 5a and Supplementary Fig. 8), confirming the immune ablative effects of free CDN. T-bet (Tbx21), an immune cell transcription factor originally described as the master regulator of Th1 cell development[24], was significantly upregulated (adjusted $p$-value <0.005) by exoSTING compared to free CDN after two doses (Fig. 5b). Tcf7 (encodes TCF1) levels were significantly decreased following 2 doses of free CDN2, suggesting a loss in stem-like progenitor CD8+ T cells[25]. This cell subset is required for response to checkpoint inhibitor therapies and this loss may underlie the lack of immunological memory response to free CDN[25]. In contrast, exoSTING treatment resulted in the upregulation of T-bet and sustained expression of Tcf7 demonstrating potent Th1 reprogramming.

Gene Set Enrichment Analysis (GSEA) showed that genes involved in innate pattern recognition receptors were upregulated by the first dose of both exoSTING and efficacious doses of free

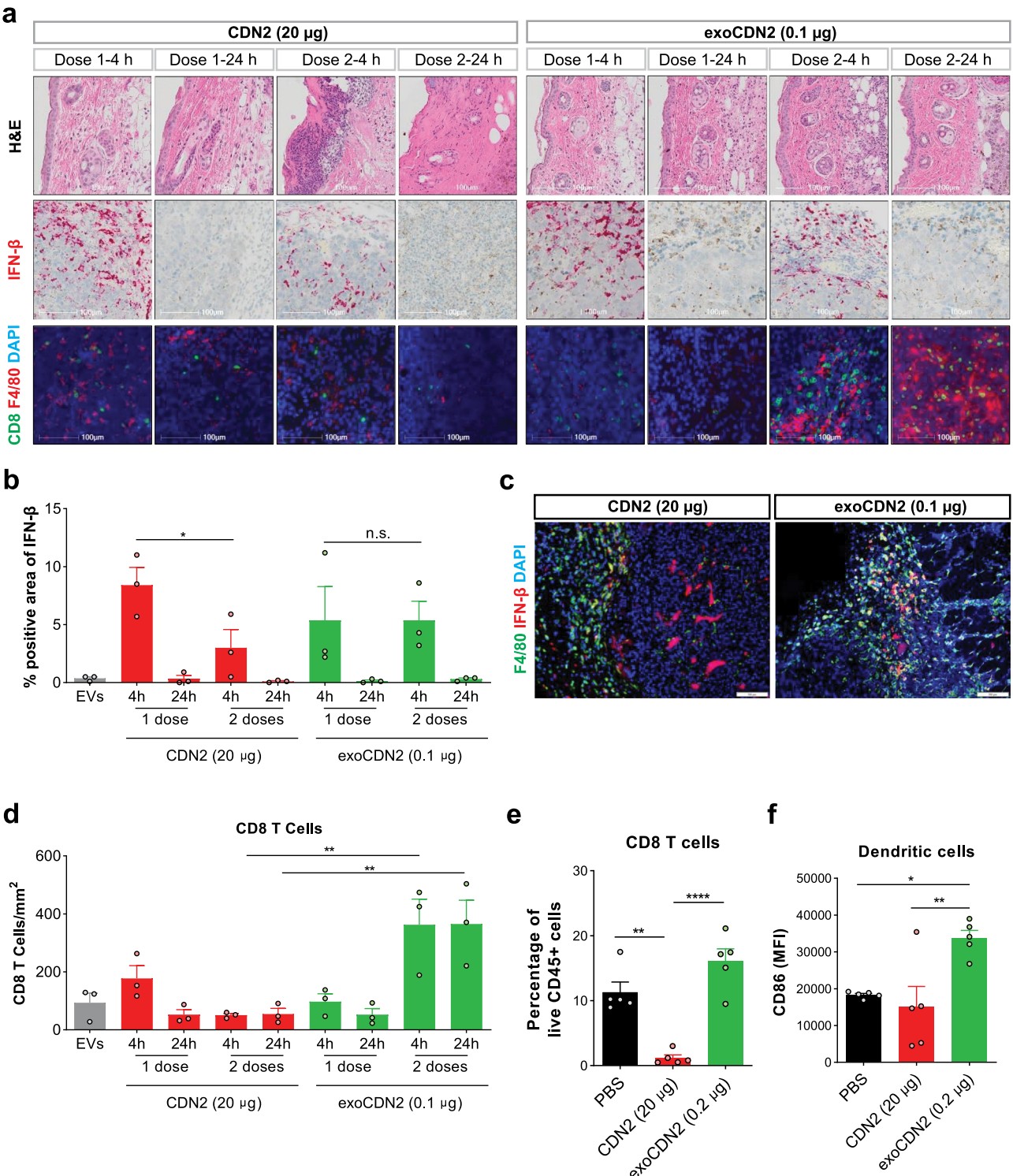

**Fig. 4 exoSTING increases T cell infiltration in tumor microenvironment without immune cell ablation. a–c** Unloaded EVs, CDN2 (20 µg), and exoCDN2 (0.1 µg) were injected intratumorally at Day 1 and Day 4 (n = 3 animals per group) into B16F10 tumors. Tumors were collected at 4 and 24 h after first dose and second doses. **a** Tumor sections were stained with H&E, or for IFN-β mRNA, CD8, and F4/80 expression. IFN-β positive area (**b**) and co-localization of F4/80 (green) and IFN-β (red) after free CDN2 (20 µg) and exoCDN2 (0.1 µg) treatment (**c**). CD8 positive cells (**d**) were measured. **e, f** Percentage of CD8 T cells in live CD45+ cells in tumors (**e**) and CD86 expression on dendritic cells (**f**) were measured by flow cytometry after two doses of PBS, CDN2 (20 µg), and exoCDN2 (0.2 µg) into B16F10 tumors (n = 5 animals per group). Data are presented as means ± s.e.m from replicate samples as indicated. *P < 0.05; **P < 0.01; ****P < 0.0001 by one-way ANOVA with Tukey's multiple comparison test; n.s. non-significant.

CDN (20 µg) (Fig. 5c) to a similar degree. ExoSTING-treated tumors (after two doses) were significantly enriched in "Th1 and Th2 activation pathway" (adjusted P-value < 1e−12), "Role of pattern recognition receptors in recognition of bacteria and viruses" (adjusted P-value < 1e−12), and "Th1 pathway" (adjusted P-value < 1e−12) transcripts (Fig. 5c). In contrast, a second dose of free CDN led to a decrease in many of these genes. This data supports the potency improvement and immune

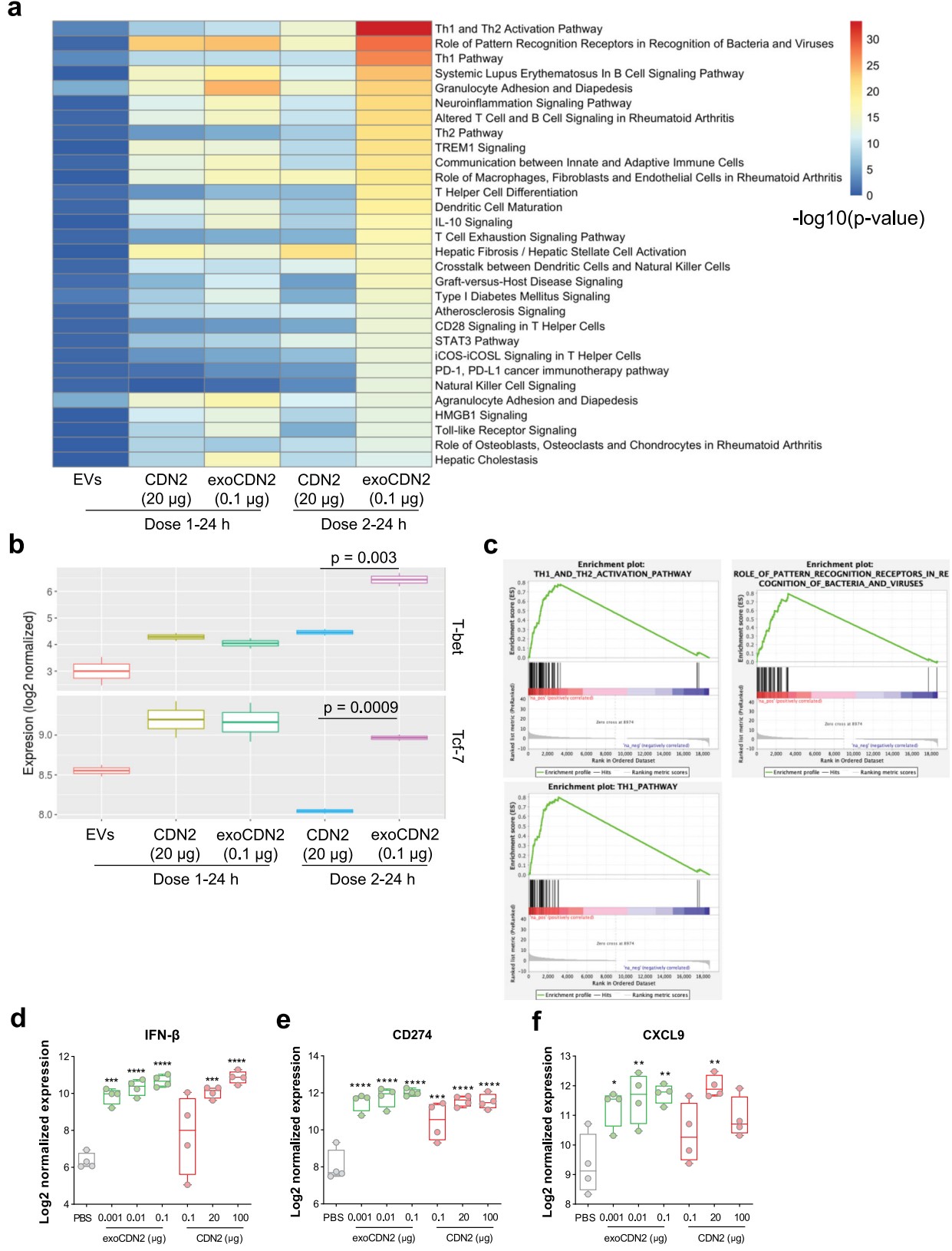

stimulatory effects of exoSTING compared to free CDN. Collectively, these data suggest that exoSTING activates differential pathways at lower doses than free CDN, and results in the activation of IFN-γ and downstream chemokines CXCL9 and CXCL10 involved in T cell recruitment. In contrast, high doses of free CDN decrease the expression of these key genes.

Using Nanostring analysis, we further evaluated the expression of immune related genes in the TME that are immediately altered (4 h) after IT injection with increasing doses of exoSTING (0.001, 0.01, and 0.1 μg) and free CDN2 (0.1, 20, and 100 μg). The immediate target gene for STING pathway activation, IFN-β, is induced in a dose-dependent manner by both exoSTING and free

**Fig. 5 exoSTING induced interferon stimulated gene signatures. a** Comparative pathway analysis of the global gene expression changes analyzed by RNA sequencing, 24 h after 1 or 2 IT injections of PBS, unloaded EVs, CDN2 (20 μg), and exoCDN2 (0.1 μg) into B16F10 tumors ($n = 2$ biological replicates per group). The data is representative of five of biological replicate samples in each treatment group. All treatments were compared to PBS treatment for gene expression. **b** Normalized expression level of Th1 transcription factors, T-bet and Tcf7. Adjusted $P$ values are indicated. **c** Gene Set Enrichment Analysis of three gene sets that are upregulated by exoCDN2. **d-f** Four hours after IT injection of PBS, CDN2 (0.1, 20, or 100 μg), and exoCDN2 (0.001, 0.01, or 0.1 μg) into B16F10 tumors, RNAs were purified from tumors ($n = 4$ animals per group) and differentially expressed genes were analyzed using NanoString technology. Relative expression of IFN-β (**d**), CD274 (**e**), and CXCL9 (**f**) measured by NanoString. Data are presented as means ± s.e.m from replicate samples as indicated. *$P < 0.05$; **$P < 0.01$; ***$P < 0.001$; ****$P < 0.0001$ by one-way ANOVA with Tukey's multiple comparison test.

CDN (Fig. 5d). Very low doses of exoSTING (0.001 μg) upregulated IFN-β mRNA by 8-fold, significantly more than a 100-fold higher dose of free CDN (0.1 μg). Similar improvements in potency were also observed in the levels of CD274 (PD-L1), a key IFN-γ-regulated gene (Fig. 5e). Expression of CXCL9 increased with exoSTING treatment (Fig. 5f), while in contrast, free CDN treatment demonstrated a bell-shaped dose response. Peak expression was observed at the 20 μg free CDN dose while the higher dose of 100 μg that was required for distal non-injected tumor control (Fig. 2d) and led to immune ablation (Fig. 2e) resulted in decreased CXCL9 production (Fig. 5f).

**exoSTING preferentially activates the STING pathway in APCs in vitro.** We assessed uptake of EVs across immune cell subtypes using EVs engineered to express luminal GFP. Analysis of cellular uptake and association with GFP containing EVs in PBMCs revealed a preferential association of EVs with APCs, monocytes (40-fold over baseline at highest dose), dendritic cells (cDC = 4-fold, pDC = 2-fold), and T cells (2-fold) (Supplementary Fig. 9a). To assess the distinct immune cell subsets activated by free CDN or exoSTING, we evaluated immune cell activation by flow cytometry with purified immune cells from PBMCs. ExoSTING treatment resulted in a dose-dependent activation of monocytes with an $EC_{50}$ of ~0.001 μM, but no activation of purified B cells, T cells, and NK cells was observed at the maximal concentrations evaluated (Fig. 6a). In contrast, free CDN activated not only monocytes ($EC_{50}$ ~0.06 μM), but at higher drug concentrations (required for anti-tumor activity) also activated T cells ($EC_{50}$ ~3.6 μM) and NK cells ($EC_{50}$ ~2.4 μM) (Fig. 6b). These data suggest that exoSTING is preferentially taken up by monocytes and activates them.

Macrophages as well as DCs represent an important class of APC in the TME. Many human tumors have been reported to be enriched in M2 immunosuppressive macrophages[26]. Both DCs and macrophages play an important role in STING agonist-mediated anti-tumor immunity[27]. To assess the effect of exoSTING on human APCs, we compared the potency of exoSTING and free CDN on DCs and M1 or M2 polarized macrophages as assessed by IFN-β production. ExoSTING induced IFN-β production at an $EC_{50}$ ~2.9 nM in purified human DCs compared to 222 nM with free CDN (Fig. 6c). In addition, exoSTING induced IFN-β production at an $EC_{50}$ ~0.05 μM in M2 polarized macrophages compared to 2.4 μM with free CDN (Fig. 6d). In contrast, exoSTING failed to induce IFN-β production at all doses tested in M1 polarized macrophages (Fig. 6e). The preferential activation of DCs and M2 macrophages by exoSTING may, at least in part, be associated with more efficient delivery of CDN to DCs and M2 macrophages. M2 polarized macrophages show ~5-fold greater uptake of EVs compared to M1 polarized macrophages (Supplementary Fig. 9b).

Next, we characterized the effect of exoSTING for free CDN on naïve or TCR-stimulated T cells. Purified T cells were treated with exoSTING or free CDN and IFN-β production was measured with and without TCR stimulation. At the higher dose levels, free CDN2 induced IFN-β production at an $EC_{50}$ ~8.2 μM (Fig. 6f) in

T cells stimulated by anti-CD3 and anti-CD28, but no IFN-β production was observed in the naïve T cells (Supplementary Fig. 9). In contrast, exoSTING did not induce IFN-β production at any dose tested from both naïve and anti-CD3 and anti-CD28 stimulated T cells (Fig. 6f and Supplementary Fig. 10). We then characterized induction of T cell death following exoSTING or free CDN2 treatment in T cells stimulated with anti-CD3 and anti-CD28. Following free CDN2 treatment, we observed a dose-dependent increase in T cell death (up to ~20% of T cells) (Fig. 6g), while no T cell death was observed at any dose of exoSTING. These results demonstrate that exoSTING can preferentially activate M2 macrophages with a significantly improved potency as compared to free CDN2 and does not induce activation of other immune cells. Importantly, exoSTING preserves the viability of TCR stimulated T cells.

**Preferential activation of APCs by exoSTING in vivo.** To determine whether the preferential activation of APCs with exoSTING in vitro would translate in vivo, we implemented a micro-dosing IT injection study using the CIVO (Comparative In Vivo Oncology) Platform paired with multiplexed immunofluorescence-based histology analysis. CIVO allows for a single tumor to be injected with multiple microdose treatments to enable comparisons of the effects of different analytes in a single mouse tumor[28]. The A20 subcutaneous B cell lymphoma model was selected to evaluate the selective uptake of EVs and cell activation parameters by histological examination of the tumors. B cells are a good surrogate to assess the "off-target" activity of pleiotropic STING activation as these cells undergo apoptosis[29]. A20 tumors were injected with exoSTING (0.02 μg), 0.02 μg or 2 μg of free CDN, and unloaded EVs as a control. Four hours after dosing, we found that 2 μg CDN treatment resulted in widespread phosphorylation of TBK1 (pTBK1) in CD19 positive B cells as well as APCs suggesting uptake of the CDN and broad activation of the STING pathway in both tumor and immune cells, whereas 0.02 μg exoSTING induced pTBK1 expression selectively in subset of immune cells that are CD19 negative (Fig. 7a, top row). Despite broad activation of pTBK1 with 2 μg of free CDN, only modest induction of IFN-β was observed (Fig. 7a, middle row; Fig. 7b). Most of IFN-β production correlated with F4/80 in the exoSTING-treated tumors, whereas IFN-β production was observed in both F4/80 positive and F4/80 negative cells in free CDN (2 μg)-treated tumors (Fig. 7d). Free CDN at a dose of 2 μg resulted in dramatic histologic changes in the injected tumor, characterized by areas of apoptotic cells around the injection site. These apoptotic scars were associated with high levels of cleaved caspase 3 (CC3). ExoSTING showed markedly less pTBK1 induction and CC3 around the injection site following injection (Fig. 7a, bottom row; Fig. 7c). These data are consistent with STING agonist-mediated apoptosis induction observed in naïve and malignant B cells[28]. Together, these data demonstrate that IT injections of high dose free CDN (which are required for anti-tumor effects in preclinical models) induced widespread STING activation, induction of pTBK1, and apoptotic cell death. In contrast, exoSTING induces preferential activation of the

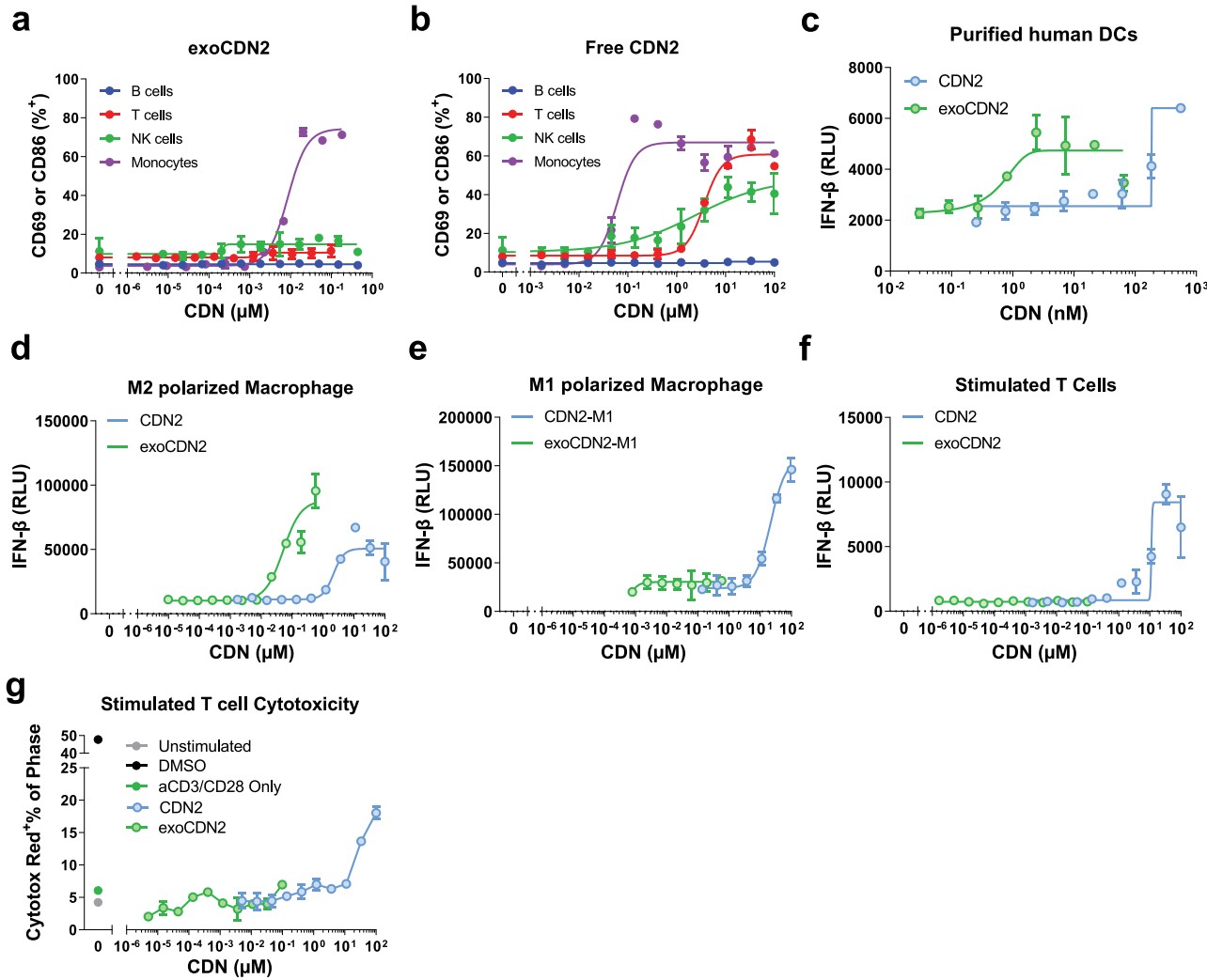

**Fig. 6 Preferentially uptake and activation of STING pathway in APCs by exoSTING. a, b** Representative dose-response curves ($n = 3$ healthy donors) of activation of purified B cells, T cells, NK cells, and Monocytes from PBMCs after treatment of exoCDN2 (**a**) or free CDN2 (**b**) ($n = 2$ biological replicates per donor). CD86 expression was assessed as a cell activation marker for monocytes, whereas CD69 was used as an activation marker for T cells, NK cells, and B cells. **c** Representative dose-response curves ($n = 4$ healthy donors) of IFN-β production in purified human DCs after treating with exoCDN2 and free CDN2 ($n = 2$ biological replicates per donor). **d, e** Representative dose-response curves ($n = 3$ healthy donors) of IFN-β production in M2 polarized human macrophages (**d**) and M1 polarized human macrophages (**e**) after treating with exoCDN2 and free CDN2 ($n = 2$ biological replicates per donor). **f, g** Representative dose-response curves ($n = 3$ healthy donors) of IFN-β production (**f**) and cytotoxicity (**g**) in stimulated T cells after treating with free CDN2 and exoCDN2 ($n = 2$ biological replicates per donor). T cells were purified from human PBMCs and stimulated with anti-CD3/anti-CD28. Data are presented as means ± s.e.m from replicate samples as indicated. RLU relative luminescent unit.

STING pathway in F4/80[+] APCs and shows dramatically reduced generalized apoptosis and tissue damage.

**Systemic administration of exoSTING results in potent anti-tumor activity in a hepatocellular carcinoma model.** The majority of the EVs injected intravenously (IV) are taken up by the liver[30]. To measure the pharmacodynamic impact of exoSTING administration via IV, we analyzed mRNA levels of IFN-β, CXCL9, and CXCL10 in the livers 4 h post-injection of unloaded EVs, exoSTING (0.2 μg) or dose matched free CDN. ExoSTING induced 10,000-fold higher levels of IFN-β in the liver compared to equivalent doses of free CDN (Fig. 8a). ExoSTING also induced substantially higher levels of CXCL9 (200-fold vs. free CDN) and CXCL10 (500-fold vs. free CDN) mRNA than a comparable amount of free CDN (Fig. 8b, c). Administration of unloaded EVs did not induce IFN-β, CXCL9, or CXCL10.

To evaluate the anti-tumor activity of exoSTING after IV administration, we used a Hepa1–6 orthotopic hepatocellular carcinoma model[31]. Hepa1–6 cells were injected into spleen to induce the tumor development in the liver. ExoSTING, dose matched free CDN or unloaded EVs were injected intravenously, and livers were collected at day 15 to assess tumor burden by liver to body weight ratio and macroscopic evaluation. ExoSTING treatment resulted in 50 % decrease in tumor burden, free CDN treatment did not decrease the tumor burden (Fig. 8d). Macroscopic evaluation to score liver lesions revealed complete remission (CR) of lesions in 3 and partial remission (PR) in 1 out of 8 mice (Fig. 8e) in the exoSTING treatment group. In contrast, no decrease in liver lesion score was observed with equivalent amount of free CDN. Representative images showed that exoSTING treated liver was similar to sham control without signs of tumors, whereas tumor growth was observed in EVs and free CDN treated liver (Fig. 8f). Collectively, these data demonstrate superior activation of STING pathway in the liver and anti-tumor activity in a hepatocellular carcinoma by exoSTING following IV administration.

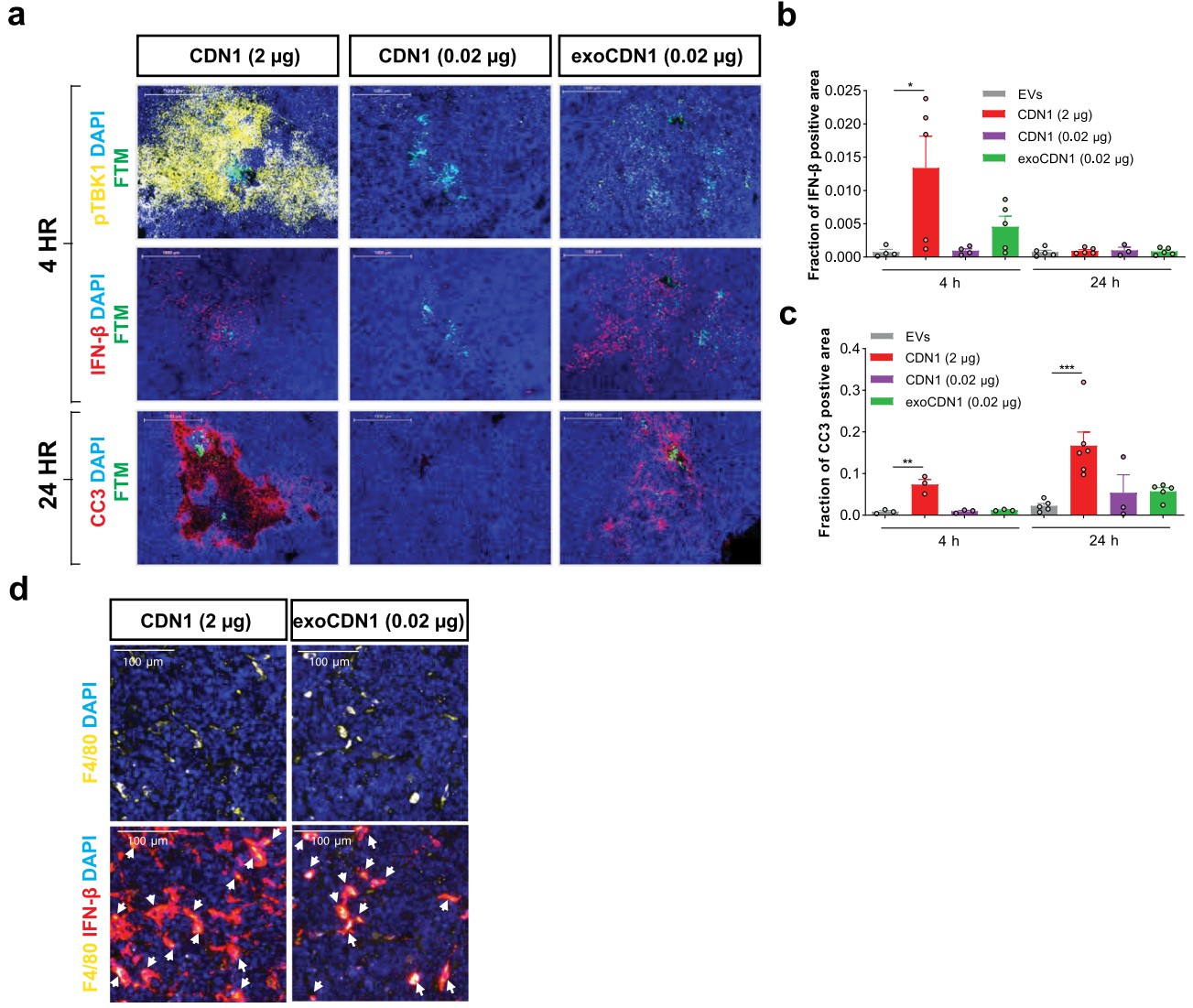

**Fig. 7 exoSTING activates the STING pathway in APCs without collateral damage in tumors. a–d** BALB/cAnNHsd mice were implanted subcutaneously with A20 B cell lymphoma cells (1 × 10⁶ cells). Unloaded EVs, CDN1 (2 or 0.02 μg), and exoCDN1 (0.02 μg) were injected into tumor ($n = 6$ animals) via the CIVO platform. **a** After 4 and 24 h, tumor sections were stained with pTBK1, IFN-β mRNA, and cleaved caspase 3 (CC3). IFN-β positive area (**b**) and CC3 positive area (**c**) were measured. Data are presented as means ± s.e.m from replicate samples as indicated. *$P < 0.05$; **$P < 0.01$; ***$P < 0.001$ by one-way ANOVA with Tukey's multiple comparison test. **d** Co-localization of F4/80 (yellow) and IFN-β (red) after free CDN (2 μg) and exoCDN1 (0.02 μg) treatment. White arrows indicate the co-localization of F4/80 and IFN-β.

**Surface glycoprotein PTGFRN enhances the potency of exoSTING.** We have identified several EV-specific proteins using unbiased proteomic analysis of highly purified EVs in a separate study. PTGFRN, a single pass transmembrane glycoprotein, was found not only to be highly abundant on EVs, but also amenable to displaying an array of structurally diverse protein cargoes on the EV surface through genetic engineering[16]. While little is known about its biological function, PTGFRN is a major component of the tetraspanin web which specifically interacts with canonical EV proteins CD9 and CD81[32]. Biochemical characterization has demonstrated a complex glycosylation status, with all 9 predicted N-linked glycosylation sites occupied[33]. DCs and macrophages are known to express several carbohydrate receptors on their surface, and EVs, via the surface glycoproteins, have been shown to bind to these sialic acid glycoprotein receptors such as Siglec-9 to facilitate internalization[34].

To understand the contribution of PTGFRN on activation of immune cells, EVs engineered to express high levels of PTGFRN⁺/⁺, normal levels of PTGFRN (WT), or PTGFRN null

EVs (PTGFRN⁻/⁻) were produced in HEK293 cells as described previously[16]. All EV populations were approximately 50–200 nm in size (Supplementary Fig. 1). These populations of EVs were next examined for their capacity activate STING pathway as measured by IFN-β production. WT, PTGFRN⁺/⁺, or PTGFRN⁻/⁻ EVs were loaded with CDN1. These CDN-loaded EVs were assayed in vitro for their potential to induce IFN-β production in PBMC cultures. EC₅₀ values and maximal IFN-β cytokine production were assessed from multiple donors. As compared to the free CDN, all EV-loaded CDN significantly enhanced potency with PTGFRN⁺/⁺ EVs resulting in the highest levels of IFN-β induction (133-fold) and PTGFRN null resulting in the lowest (29-fold) (Supplementary Fig. 11a, b). In addition, we compared the in vivo activity of different EVs (WT, PTGFRN⁻/⁻ and PTGFRN⁺/⁺) loaded with equal amounts of CDN in the B16F10 tumor model. We observed that anti-tumor activity was correlated with PTGFRN density on EVs with PTGFRN⁻/⁻ EVs having minimal anti-tumor activity (Supplementary Fig. 11c). We next evaluated the possible direct

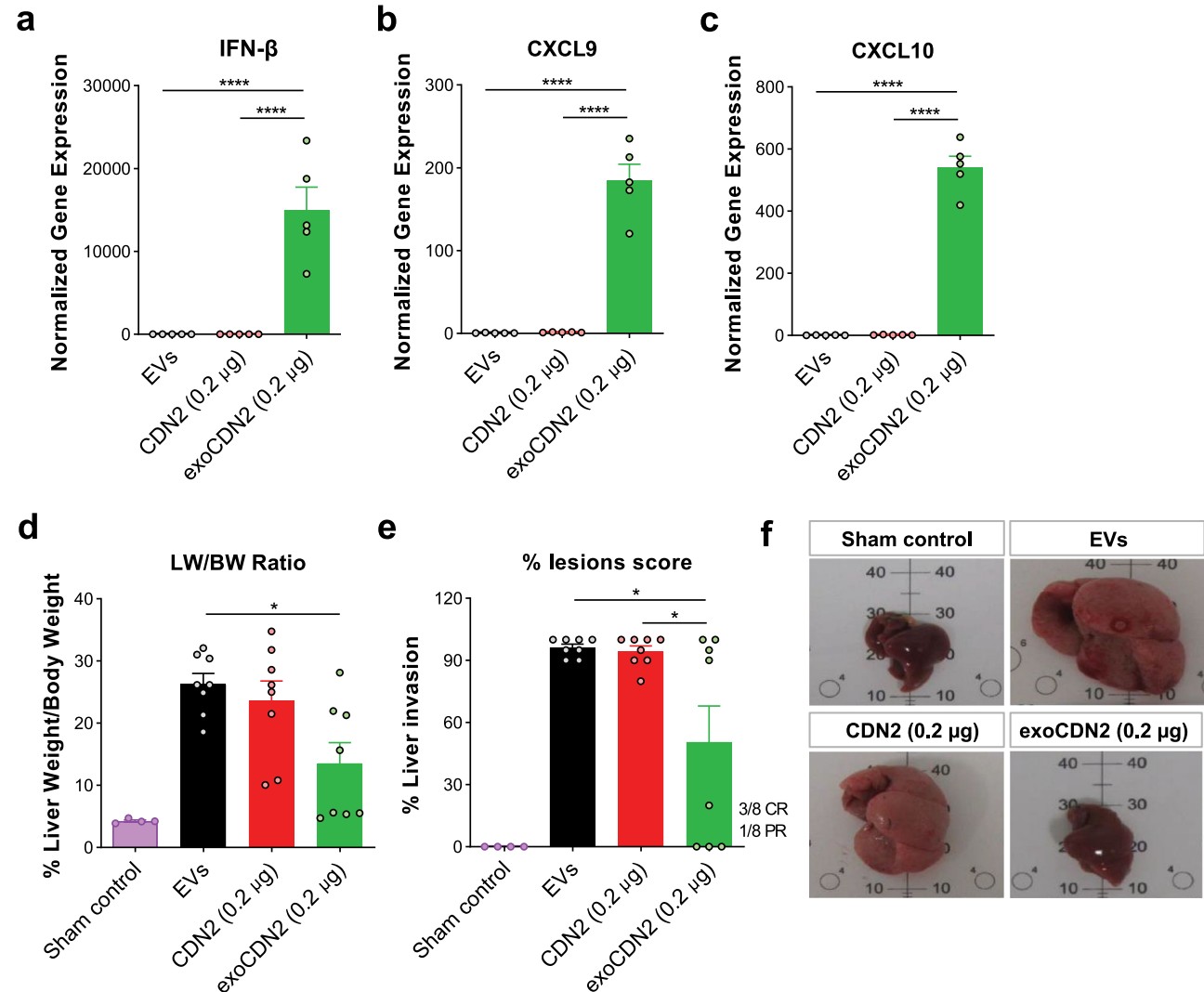

**Fig. 8 Systemic administration of exoSTING enhances pharmacodynamic response and anti-tumor activity of a CDN. a–c** Four hours after IV injection, RNAs were purified from livers ($n = 5$ animals per group). Relative expression of IFN-β (**a**), CXCL9 (**b**), and CXCL10 (**c**) genes was measured by RT-qPCR, normalized against the housekeeping gene *RPS13*. **d–f** Orthotopic hepatocellular carcinoma was induced by injecting $1.5 \times 10^6$ Hepa1–6 cells into spleen. Unloaded EVs, CDN2 (0.2 μg), and exoCDN2 (0.2 μg) were injected intravenously at Day 4, 7, and 10 after cell injection ($n = 8$ animals per group). At Day 15, mice were sacrificed, and livers were collected. Liver weight (LW) over body weight (BW) was calculated (**d**) and % lesions scores were evaluated (**e**). Representative liver images were taken (**f**). Data are presented as means ± s.e.m from replicate samples as indicated. *$P < 0.05$; ****$P < 0.0001$ by one-way ANOVA with Tukey's multiple comparison test.

association of PTGFRN with to the sialic acid binding glycoprotein receptors using Bio-Layer Interferometry (BLI). Binding of PTGFRN to Siglec-9, Siglec-10, and Siglec-14 (KD $7 \times 10^{-7}$ to $4 \times 10^{-6}$ M), which are enriched in APCs was observed in contrast PTGFRN did not bind to Siglec-2, Siglec-4, and Siglec-8 (Supplementary Fig. 12). Although the precise mechanism of CDN delivery to the cytosol of the APC and the role of PTGFRN in mediating exoSTING potency is yet to be established, these data suggest that glycoprotein PTGFRN on EVs may play a role in maximizing activation of immune cells and in vivo anti-tumor activity, which may be contributed by cellular tropism and immune cell signaling activity of PTGFRN[17,34].

## Discussion

Several STING agonist drug candidates are currently being assessed in human clinical trials of various malignancies[35], however early single-agent human results have not been as promising as predicted by preclinical models. Recent observations

regarding the PK/PD relationships with the CDN class of STING agonists may explain the apparent discordance between preclinical and clinical activity profiles[10–12]. CDN molecules exhibit a paradoxical bell-shaped dose response curve in in vivo preclinical tumor models[12] where immune stimulation is seen at a narrow range, but with increasing concentrations the anti-tumor effect is abrogated. This report substantiates high-dose immune ablation with a 3′-3′ CDN (referred as CDN2, Figs. 2, 4, and 7), similar to results described recently for the 2′-3′ CDN ADU-S100[12]. Due to low cellular permeability and poor tissue retention, high concentrations of free CDNs are required for anti-tumor activity and these lead to progressive loss of tumor-infiltrating immune cells, especially T cells and APCs (Fig. 4). This is accompanied by decreased expression of CXCL9 and a key transcription factor TCF-1 in the tumor following free CDN2 administration (Fig. 5). The molecular changes observed following free CDN administration are consistent with the ablation of systemic T cell-mediated immunity and the absence of immunological memory (Fig. 2e). Another consistent observation

(preclinically[36] and clinically[10,11]) with CDNs delivered by IT injection is the high degree of extravasation of the CDN and short residence time in the tumor (Fig. 3b). The consequence of this rapid extravasation is transient exposure of the tumor to the CDN and the necessity to increase the injected dose, further exacerbating the immune ablation and resulting in systemic inflammation due to STING activation outside the TME.

These limitations of CDNs (poor cytosolic delivery, tumor extravasation, and nonselective delivery) could potentially be addressed by nanotechnologies[37]. It has been reported that liposomes[38] or polymersomes[39] enhanced the anti-cancer activity of CDN with increased T cell responses. EVs are natural mediators of communication between tumor cells and immune cells and have been shown to transmit tumor-derived dsDNA to engage the STING pathway and provoke immune surveillance in the TME[14,15]. Our goal was to address the limitations of CDN by using EVs as a delivery vehicle that would specifically activate the STING pathway in APCs while avoiding T cell ablation[12]. Our studies demonstrate the advantages of improved potency and preferential activation of APCs with exoSTING leading to improved anti-tumor activity and protective immunity compared to free CDN.

In vitro and in vivo studies demonstrated a 100–200-fold improvement in potency by exoSTING mediated CDN delivery. As demonstrated previously[9,12], this efficacy was dependent on STING expression in host immune cells (Supplementary Fig. 5e) and CD8$^+$ T cells (Fig. 2f). Consistent with previously published studies with other CDNs[12], our results confirm the need for high doses of free CDN (20–100 µg) to promote local tumor control in the injected lesion, likely due to direct cytotoxic effects of high dose CDN on the tumor cells. Non-injected contralateral tumor control was also observed with exoSTING in the dual flank tumor model (Fig. 2d), while high levels of free CDN (100 µg) were required for contralateral tumor growth control (Fig. 2d). ExoSTING, but not free CDN, resulted in immunological anti-tumor memory as demonstrated protection from tumor re-challenge of exoSTING treated animals (Fig. 2e). The immune ablative effects of the therapeutically active doses of free CDN were confirmed by loss of CD8$^+$ T cells, macrophages, and DCs in the injected tumor with repetitive dosing (Fig. 4 and Supplementary Fig. 7). The ablation of immune cells with free CDN was associated with the bell-shaped dose response curve for CXCL9 and IFN-γ mRNA expression, as well as reduced expression of TCF-1 (Fig. 5). CXCL9 is predominantly made by the XCR1$^+$ BATF3 lineage DCs[40], which are important for establishing STING-mediated systemic T cell responses. TCF-1 is required for maintenance of the stem-like CD8 effector pool in the TME[25] and exoSTING preserves and expands this population in injected tumors (Fig. 5b). Taken together these data suggest that the immune cells responsible for establishing systemic immunity are affected negatively by high doses of free CDN treatment. In vitro studies further confirmed preferential activation of APCs with exoSTING, resulting in activation of STING pathway in monocytes, DCs and M2 macrophages and ~100-fold increased potency (Fig. 6c, d). In contrast, consistent with previous reports[41], free CDN treatment resulted in STING activation as measured by IFN-β production and cell death of TCR stimulated T cells, while exoSTING did not induce STING activation or T cell death. These results are consistent with T cell preservation and the improved anti-tumor immunity observed with exoSTING.

HEK293 is a suitable host cell line for scalable, GMP-compatible production of engineered EVs[19]. Unloaded EVs derived from HEK293 were largely immune silent and did not activate IFN-β production in vitro (Supplementary Fig. 3a) or modulate the transcriptome in vivo (Fig. 5 and Supplementary Fig. 6a) and did not control tumor growth (Fig. 1d) making it an ideal vehicle for drug delivery. VSV-G a fusigenic peptide was required for efficient cGAS, a 62 kD cytosolic protein, mediated STING activation[42]. Here, we show robust activation of STING pathway in the absence of VSV-G peptide. Fusogenic peptides like VSV-G may be required for endosomal escape of a large protein like cGAS for efficient cytosolic delivery. In this work, we show that the presence of complex EV surface proteins may provide a unique advantage in preferential uptake and activation of APCs. We have identified association of these EV surface proteins, PTGFRN with Siglec-9, 10, and 14. EVs were preferentially taken-up by APCs in PBMCs, whereas liposomes were taken-up by all cell types without selectivity (Supplementary Fig. 13a). In addition, liposomes loaded with CDNs and formulated as previously described[38] induced dose-dependent reduction of macrophage viability in vitro, while exoSTING maintained the viability of these APC (Supplementary Fig. 13b). Liposome mediated toxicity has been well documented[43]. These results highlight the advantage of EV mediated preferential delivery of CDN and preservation of viability for APCs. Furthermore, in vitro studies confirmed the lack of delivery of exoSTING to T cells, leading to preservation of this crucial immune effector population. These results confirm and extend the previously reported immune ablation by free CDN[12], and illustrate the selective and superior activity of exoSTING to promote systemic, antigen-specific immunity.

ExoSTING leverages the natural properties of EVs to communicate between tumor cells and APCs in the TME and overcomes major limitations of free STING agonists. ExoSTING enables effective intracellular delivery of CDN STING agonists, prolongs tumor retention of the drug, and selectively targets APCs in the TME. These properties significantly improve potency, limit systemic inflammation, and prevent immune ablation, achieving systemic antitumor immunity and avoiding the bell-shaped pharmacology seen with free CDN STING agonists. The improved potency and wider therapeutic window of exoSTING should enable optimized dosing strategies in the clinic through precise delivery of CDNs.

## Methods

**Cell lines and culture**. B16F10 (ATCC CRL-6575), EG7-OVA (ATCC CRL-2113), CT26.WT (ATCC CRL-2638), and Hepa1–6 (ATCC CRL-1830) were purchased from ATCC. B16F10 and Hepa1–6 were cultured in DMEM media (Gibco). CT26.wt were cultured in RPMI media (Gibco). All media were supplemented with 10% fetal bovine serum and 1× penicillin/streptomycin (Invitrogen). EG7-OVA was cultured in RPMI media supplemented with 10% fetal bovine serum, 10 mM HEPES, 1 mM sodium pyruvate, 2 mM L-glutamine, 0.05 mM 2-mercaptoethanol, and 0.4 mg/mL G418. All cells tested negative for mycoplasma.

**Transfection and stable cell line selection**. HEK293 cells adapted for suspension were grown in CDM4PERMAb media supplemented with 4 mM L-glutamine (GE Healthcare). DNA cassettes encoding PTGFRN with and without a C-terminal GFP tag were cloned downstream of a CMV promoter and introduced into the HEK cells via electroporation or PEI (Polysciences) mediated transfection. Selection of stable cell lines was achieved by adding puromycin or neomycin and routinely passaged using the cell pools returned to a viability suitable for cryo-preservation (>90%). A clonal cell line expressing PTGFRN alone was selected by two rounds of limited dilution and PTGFRN overexpression was confirmed by SDS-PAGE and Western blot. A HEK293 PTGFRN knockout pool was generated using CRISPR/Cas9 editing with guide RNAs targeting regions in exons 2 and 9 according to the manufacturer's protocol (ThermoFisher). A clonal cell line was selected by limited dilution and PTGFRN knockout was confirmed by genotyping and Western blot.

**EV isolation**. EVs were produced and isolated from WT, PTGFRN overexpressed, and PTGFRN knock-out HEK293 cells as described previously[16]. Chemically defined media (CDM4PERMAb) was inoculated at 0.3E+06 viable cells per mL at the 10–25 L scale using WAVE bioreactors (GE Healthcare) maintained at 37 °C and 8.0% CO$_2$. Cells were grown for 9 days with cell density and viability measured daily on a Vi-CELL XR cell counter (Beckman Coulter). At the termination of the culture, cells were removed by centrifugation at 6000×$g$ for 15 min. The cell pellet

was discarded, and the clarified conditioned media was filtered using Sartopore 0.8/0.45 μm MidiCaps (Sartorius), supplemented with MgCl₂ to a final concentration of 1 mM, and treated with 20 U/mL Benzonase (Millipore) overnight at room temperature with gentle agitation. Nuclease treated media was next concentrated 10× by tangential flow filtration using a SARTOFLOW benchtop system equipped with Pellicon 2 mini 1000 NMWL polyethersulfone (PES) ultrafiltration cassettes (Millipore). Concentrated media was loaded into 100 mL Quick-Seal Ultra-Clear tubes and centrifuged for 60 min at $133,900 \times g$ at 4 °C in a 45 Ti fixed-angle rotor in an Optima XE ultracentrifuge (Beckman Coulter). The crude EV-containing pellets were resuspended in minimal volumes of sterile PBS for further processing.

The resuspended crude pellets were adjusted to a final volume of 3 mL with sterile PBS and mixed with 9 mL of 60% iodixanol solution (OptiPrep, Sigma), resulting in a final concentration of 45% iodixanol. This mixture was transferred to a 38 mL UltraClear tube (Beckman Coulter). Successive layers of lower density iodixanol solutions were carefully pipetted on top of the 45% layer: 9 mL of 30%, 6 mL of 23%, 6 mL of 18%, and 3 mL of PBS. These lower density solutions were prepared by diluting OptiPrep with a homogenization buffer (250 mM sucrose, 10 mM Tris-HCl, 1 mM EDTA, pH 7.4) to achieve the final indicated iodixanol vol/vol percentages. A density gradient was achieved by centrifuging at $150,000 \times g$ for 16 h at 4 °C in a swinging-bucket SW 32 Ti rotor (Beckman Coulter). EVs were isolated from the interface between the PBS and 18% iodixanol layer by careful pipetting and transferred to a clean 38 mL UltraClear tube. An additional low speed spin at $20,000 \times g$ for 30 min at 4 °C was used to remove any contaminating actin and actin-binding protein species. The supernatant was filtered using a 0.22 μm PVDF sterile filter, transferred to a clean 38 mL UltraClear tube, and centrifuged at $133,900 \times g$ for 3 h at 4 °C. The final purified EV pellet was resuspended in a minimal amount of sterile PBS, characterized, aliquoted, and frozen at −80 °C for long term storage.

**Western blot.** Approximately 3E10 EVs per sample were diluted in reducing Laemmli buffer, denatured at 95 °C for 10 min, and loaded into precast 4–20% TGX Stain-free gels (Bio-Rad). Separated protein was transferred to a PVDF membrane using a Trans-Blot Turbo transfer system (Bio-Rad), and blocked in 1% casein for 1 h. Primary antibodies were diluted in blocking buffer and incubated with membranes for 1–3 h; proteins of interest were detected using HRP-conjugated secondary antibodies and chemiluminescent substrate. Primary and secondary antibody information is listed in Supplementary Table 2.

**Transmission electron microscopy.** EV samples were diluted to 2E11 EVs per mL and incubated for 1 min on a 200-mesh Formvar™ and carbon-coated copper grid (Ted Pella). The grids were rinsed with water and the excess solution was wicked away, and then stained with a 1% solution of uranyl acetate for 30 s. Excess staining solution was wicked away, and the grids were allowed to dry prior to imaging with a Philips CM12 transmission electron microscope operating at 80 kV.

**Nanoparticle tracking analysis.** EV concentration was measured by using nano-tracking analysis on a NanoSight NS3000 (Malvern Panalytical, Westborough, MA, USA). Video images were recorded for 30 s with camera level 14 and particles were analyzed using the nanoparticle tracking analysis (NTA) software (version 3.2) with detection threshold 5. Measurements were performed in triplicate for each sample.

**STING agonist loading into EVs.** One micromolar of CDNs, MR SS-2 CDA (MedChem Express, Cat # HY-12885B) or cAIM(PS)2 Difluor (Rp/Sp) (Invivogen, Cat # tlrl-nacairs) was mixed with EVs and incubated for 24 h at 37 °C. EVs were ultracentrifuged at $100,00 \times g$ (TLA120.2, Beckman) for 20 min by using Optima MAX XP (Beckman). Supernatant was decanted, pellets washed with PBS once, and resuspended in PBS. Resuspended EVs were kept in −80 °C until use.

**LC-MS/MS quantitation of a STING agonist.** STING agonist standard curves were prepared by serial dilution in phosphate buffer containing 1E+11/mL EVs such that all standards contained an equal concentration of EVs. All samples were appropriately diluted, so the final concentration of EVs was equal to that of the standards, 1E+11/mL EVs. All standards and samples were then transferred to HPLC vials and diluted 3:1 with EV lysis buffer (60 mM Tris, 400 mM GdmCl, 100 mM EDTA, 20 mM TCEP, and 1.0% Triton X-100), followed by the addition of 2.0 μg of Proteinase K enzyme (Dako, reference S3004). All vials were then capped, vortexed to mix, and incubated at 55 °C for 60 min. Following incubation, all HPLC vials were allowed to cool to room temperature and were held at 4–8 °C until analysis.

The concentration of STING agonist was also measured in plasma or tissue homogenate samples. Samples delivered in these matrices were compared to a standard curve of the identical STING agonist prepared in plasma matching the species, strain, gender, and anticoagulant used for the in-life portion of the study. All standards and samples were subjected to a protein precipitation step by adding five volumes of acetonitrile containing a second cyclic dinucleotide used as an internal standard. They were then vortexed vigorously to mix and centrifuged to pellet precipitated material. The supernatants were then collected and evaporated

under nitrogen gas, reconstituted in water and 0.1% formic acid, and then held at 4–8 °C until analysis.

Standards and samples were injected neat into an ACQUITY UPLC I-Class System (Waters Corporation). Separation of analytes was performed using an ACQUITY UPLC HSS T3 analytical column (2.1 × 50 mm, 1.8 μm particle size, 100 Å pore size; Waters Corporation) and a gradient of mobile phase A (MPA: water, 0.1% formic acid) and mobile phase B (MPB: acetonitrile, 0.1% formic acid) at a flowrate of 500 μL/min. The gradient began at 0% MPB, which was held for 1 min to load and desalt the STING agonist analyte. The percentage MPB then increased from 0 to 95% over 1.5 min to elute the analyte. The percentage MPB was held at 95% for 1.25 min, decreased from 95 to 0% over 0.25 min, and then held at 0% for 1 min to re-equilibrate the column. The total runtime for the method was 5 min, and LC flow was only directed into the MS between 1.0 and 2.5 min. Samples were typically injected in duplicate with blank injections performed between unique analytical samples.

Mass analyses were performed with a Xevo G2-XS QTof (Waters Corporation) quadrupole time-of-flight mass spectrometer with an electrospray ionization (ESI) probe, and source parameters were optimized for the LC flow rate of 500 μL/min. Analyses were performed using Tof-MRM mode, negative polarity, and sensitivity analyzer mode. Time-of-flight data (continuum format) were acquired across the $m/z$ range from 100 to 600 Da with a scan time of 0.1 s. Multiple reaction monitoring (MRM) data used for quantitation were acquired using a precursor $m/z$ of 346.5 Da and a fragment $m/z$ of 557.97 Da. The concentration of STING agonist in a given sample was determined by comparing the STING agonist peak area in that sample to STING agonist peak areas generated by standards. In cases where an internal standard was used, the concentration of STING agonist was determined by comparing the ratio of analyte response and internal standard response in a given sample to the ratios measured in the standards.

**In vitro human PBMC assay.** Healthy human whole bloods were purchased from Research Blood Components LLC. All donors filled out their standard medical questionnaire and agreed to their donor informed consent form before blood donation. PBMCs were isolated from whole blood using SepMate tubes (STEM-CELL Technologies). Cells were plated in round-bottom 96-well plates at 200,000 cells per well in RPMI supplemented with 10% fetal bovine serum. ExoSTING or free CDNs were added to the wells in a final volume of 200 μL and incubated overnight at 37 °C and 5% CO₂. The next day, the supernatant was harvested and analyzed for human IFN-β using an AlphaLISA kit according to the manufacturer's protocol (Perkin Elmer). The cells were pelleted, washed, and stained for flow cytometry. Expression of the activation marker CD86 on CD11c+ dendritic cells and CD14+ monocytes were assessed. Flow cytometry analysis was completed either on a SA3800 Spectral Cell Analyzer (Sony) or Beckman Coulter CytoFLEX LX cytometer. EC₅₀ values were analyzed by using GraphPad Prism 8. Antibody information is listed in Supplementary Table 2.

**Generation of M1 and M2 macrophages from monocytes.** Monocytes were isolated from whole blood by negative selection using RosetteSep™ Human Monocyte Enrichment Cocktail (STEMCELL Technologies). Monocytes were plated at 50,000 cells per well in flat-bottomed 96-well plate, in 200 μL RPMI supplemented with 10% fetal bovine serum and 40 ng/mL human recombinant M-CSF (BioLegend). Cells were incubated for 5 days at 37 °C and 5% CO₂, with one change of medium containing M-CSF on day 3. For M1 macrophages, cells were treated with 40 ng/mL IFN-γ on day 5 and allowed to differentiate for at least 24 h. For M2 macrophages, cells were treated with IL-4, TGF-β, IL-10, each at 20 ng/mL, and allowed to differentiate for at least 24 h. Spent media was removed and exoSTING or free CDN were added to the wells in a final volume of 200 μL and incubated overnight at 370 °C and 5% CO₂. The next day, the supernatant was harvested and analyzed for human IFN-β using an AlphaLISA kit according to the manufacturer's protocol (Perkin Elmer). For uptake, EVs were labeled with pHrodoRed NHS dye according to the manufacturer's protocol (ThermoFisher). Labeled EVs were incubated with cells and fluorescent signals were measured with Incucyte.

**Animals.** Five to six-week-old female C57BL/6 and BALB/c mice were purchased from Taconic and The Jackson Laboratory, respectively. All animals were maintained and treated at the animal care facility of Codiak Biosciences in accordance with the regulations and guidelines of the Institutional Animal Care and Use Committee (CB2017-001). For the Hepa1-6 study, five to six-week-old female C57BL/6 mice were purchased from Janvier Labs (Le Genest St Isle, France). Animal housing and experimental procedures were conducted according to the French and European Regulations and the National Research Council Guide for the Care and Use of Laboratory Animals and Institutional Animal Care and Use Committee of Oncodesign (Oncomet) approved by French authorities (CNREEA agreement N° 91).

**In vivo mouse tumor models and treatment.** To establish syngeneic tumor models, B16F10 ($1 \times 10^6$ cells to C57BL/6), CT26 ($5 \times 10^5$ cells to BALB/c), and EG7-OVA ($1 \times 10^6$ cells to C57BL/6) were injected subcutaneously to the right flank of mice. For lung metastasis, B16F10 ($1 \times 10^5$ cells) were injected

intravenously to B16F10 tumor bearing mice, 4 days after subcutaneous injection. For dual flank model, $1 \times 10^6$ and $5 \times 10^5$ B16F10 cells were injected subcutaneously to the right and left flank of mice, respectively. When tumor reaches an average volume of 50–100 mm³, the mice were randomized into several groups according to the experimental protocol. Tumor volume (mm³) was calculated as (width)² × (length) × 0.5. IT dosing on the right tumors was performed three times with 3 days interval. To establish a mouse orthotopic hepatocarcinoma model, Hepa1-6 cells ($1.5 \times 10^6$) in HBSS medium were injected into spleen of C57BL/6 mice and the spleen was excised subsequently. IV dosing was performed three times with 3 days interval from Day 4 after cell inoculation. Isotype control antibody (10 mg/kg, BioLegend), Anti-CD8 (10 mg/kg, BioLegend, Clone 53–6.7), Anti-NK1.1 (10 mg/kg, BioLegend, Clone PK136), Anti-CSF1R (10 mg/kg, BioXcell, Clone AFS98), Anti-PD1 (10 mg/kg, BioLegend, Clone RMP1–14), and Anti-CTLA4 (5 mg/kg, BioLegend, Clone 9H10) were administered intraperitoneally.

**Gene expression analysis by qPCR.** Subcutaneously injected B16F10 tumors with ~100 mm³ were administered intratumorally with PBS, free CDN at 20 or 0.2 µg, and exoSTING (0.2 µg). After injection of samples, tumors, tumor-draining lymph nodes, spleens were collected and kept in RNA later solution for overnight at 4 °C. In another study, PBS, free CDN (0.2 µg), and exoSTING (0.2 µg) were injected intravenously, and livers were collected at 4 h after injection. For gene expression analysis, tissues were lysed Trizol solution with mechanical disruption. RNAs were isolated by using RNeasy Lipid Tissue Mini Kit (Thermo Fisher Scientific), according to manufacturer's instructions. cDNA was synthesized by using Superscript IV VILO Master Mix (Thermo Fisher Scientific). Target mRNA expression was measured by using TaqMan™ Universal PCR Master Mix (Thermo Fisher Scientific) and QuantStudio™ 3 & 5 Real-Time PCR System (Thermo Fisher Scientific). Ct values for all target genes were normalized to Ct values of the housekeeping gene RPS13. TaqMan™ gene expression probes were: Mm00439552_s1 for mouse Ifnb1; Mm00434946_m1 for mouse Cxcl9; Mm00445235_m1 for mouse Cxcl10; Mm01168134_m1 for mouse Ifng; Mm00850011_g1 for mouse Rps13.

**NanoString analysis.** Fifty nanogram of total RNA was incubated with a Reporter Code set and a Capture Probe set from a nCounter® Mouse Myeloid Innate Immunity Panel (for mouse tumor samples) for overnight at 65 °C. Then, mixture was injected into a nCounter® SPRINT Cartridge and analyzed by a nCounter® SPRINT Profiler. Raw files were analyzed by nSolver Analysis Software 4.0 and normalized gene expression levels were obtained. Differential expression analysis was performed with Welch's t-test. The genes with P-value < 0.05 or fold change > 2 were defined as differentially expressed genes. Gene set enrichment test (GSEA) was performed using the genes in the NanoString panel with ranked scores that indicate the dosage/time dependency or expression fold changes. The pathways/gene sets with FDR < 0.25 were defined as significantly enriched.

**RNA sequencing and data analysis.** cDNA libraries were prepared from mRNA in the tumor issue of treated mice and sequenced using an Illumina HiSeq, 2 × 150 bp. RNA-seq FASTQ files were processed using the customized bulk RNA-seq data analysis pipeline in OmicSoft ArraySuite, version 10.1. Specifically, raw reads were first filtered based on quality control (QC) and then aligned to the reference genome (Genome Reference Consortium Mouse Build 38/GRCm38) using OSA. After alignment, gene expression levels (raw read counts and FPKMs) were quantified by the RSEM algorithm33 with the mouse gene model Ensembl.R88. Gene counts were normalized by library size factors with the R package DESeq2. Differential expression analysis was performed using DESeq2. We considered genes as differentially expressed if the adjusted P-value is less than 0.05 and the expression ratio is >2×. Pathway analysis was performed with Ingenuity Pathway Analysis (IPA) (QIAGEN Inc., https://www.qiagenbioinformatics.com/products/ingenuitypathway-analysis) and Gene Set Enrichment Analysis (GSEA).

**Pharmacokinetics analysis of STING agonist in tumors.** Subcutaneously injected B16F10 tumors with ~100 mm³ were administered intratumorally with PBS, free CDN at 0.3 and 30 µg, and exoSTING (0.3 µg). 5, 30 min, 2, 6, 24, and 48 h after injection, tumors were collected and immediately freeze-down by using dry-ice. Tumors were thawed and weighted. Six volume of normal mouse plasma, purchased from BioIVT, was added to tumor and homogenized with a bead rupture. The concentration of STING agonists was measured by LC-MS/MS as described above.

**Serum cytokine measurement.** Blood was obtained by cardiac puncture from mice and serum was isolated by centrifuging at 10,000× g for 90 s. Thirteen cytokines including IFN-β, TNF-α, IL-6, and MCP-1 were analyzed by using LEGENDplex™ Mouse Inflammation Panel (13-plex) (BioLegend), according to a Manufacturer's instruction. Briefly, serum was mixed with equal volume of assay buffer, added to a LEGENDplex™ Mouse Inflammation Panel plate, and incubated for 2 h at room temperature. A plate was washed twice, and detection antibodies were added and incubated for 1 h at room temperature. PE conjugated streptavidin was incubated for 30 min, and samples were analyzed by flow cytometry.

**ELISPOT.** Spleens were harvested and dissociated into single cell suspension of splenocytes by manually grinding the spleen over a 40 µm filter (Falcon) and red blood cells lysed using ACK Lysing Buffer (Thermo Fisher). Cells were washed with PBS and resuspended in RPMI 1640 with L-Glutamine (Thermo Fisher Scientific), 10% fetal bovine serum (Thermo Fisher Scientific), and 1% Antibiotic–Antimycotic (Thermo Fisher Scientific). Cytokine analysis was performed using the mouse IFN-γ ELISpotPLUS Kit (Mabtech), according Manufacturer's protocol. Briefly, plates were blocked with serum-containing culture media and stimuli and cell suspension (400,000 per well) added. B16 antigen pool used was a mixture of GP100 (KVPRNQDWL), TRP-2 (SVYDFFVWL), and TYR (YMDGTMSQV) (AnaSpec), with each having a final assay concentration of 2.5 µg/mL. Plates were wrapped in foil and incubated for 18 h at 37 °C, 5% CO₂. Following stimulation, the cells were removed, plate washed, and 1 µg/mL detection antibody added for 2 h at room temperature. Wash was repeated and 1× Streptavidin-HRP added and incubated for 1 h at room temperature. Finally, plates were washed and TMB substrate added, incubated for 4 min in the dark for spot development, then washed out using tap water. Plates were allowed to dry and counted (ZellNet Consulting).

**CIVO assay.** A20 cells (ATCC) were cultured in RPMI 1640 with L-Glutamine (Thermo Fisher Scientific), 10% fetal bovine serum (Thermo Fisher Scientific), and 50 nM β-Mercaptoethanol at 37 °C and 5% CO₂. All experiments in mice were approved by IACUC Board of Presage Biosciences, Seattle, WA (Protocol number PR-001) and were performed at Presage in accordance with relevant guidelines and regulations. For generating A20 allografts, female BALB/cAnNHsd mice (Envigo) were inoculated with $1 \times 10^6$ A20 cells. CIVO IT microinjections were performed as described previously[28]. Briefly, mice (n = 6 per time point, 4 and 24 h) were enrolled in microinjection studies when implanted tumors reached the following approximate dimensions: 14 mm (length), 10 mm (width) and 7 mm (depth). The CIVO device was configured with 6 thirty-gauge injection needles with a total volume delivery of 2 µL. Presage's fluorescent tracking marker (FTM, 5% by volume) was added to the injection contents for spatial orientation. At 4 and 24 h following CIVO microinjections, mice were euthanized using CO₂ inhalation for biomarker analyses.

**EV uptake.** PBMCs were isolated from whole blood using SepMate tubes (STEMCELL Technologies). Cells were plated in round-bottom 96-well plates at 200,000 cells per well in RPMI supplemented with 10% fetal bovine serum. Different EVs were added to the wells in a final volume of 200 µL and incubated overnight at 37 °C and 5% CO₂. The next day, the cells were pelleted, washed, and stained for flow cytometry. Level of the GFP was assessed, along with different population markers (CD3—T Cells, CD19—B Cells, CD16/56—NK Cells, CD14—Monocytes, CD123—pDCs, CD11c—cDCs). Flow cytometry analysis was completed on a SA3800 Spectral Cell Analyzer (Sony).

**Examination of TIL in vivo by flow cytometry.** ExoCDN2 (0.2 µg) or free CDN2 (0.2 and 20 µg) were injected into B16F10 tumors of approximately 50–100 mm³ and dosed again 3 days after. Twenty-four hours after injection, tumors were excised and processed using a mouse tumor dissociation kit and gentle MACS system (Miltenyi Biotec) per the manufacturer's instruction. Single cell suspensions were washed twice with PBS and stained with fluorescently conjugated antibodies. Immune infiltrate of tumors were characterized by flow cytometry through CD45 expression (30-F11) and analyzed for CD8+ T cells by expression of CD3e+ (145-2C11)TCRβ+ (H57-597)CD8+ (53–6.7), macrophages by expression of CD11b+ (M1-70)F4/80+ (6F12)I/A-I/E+(M5/114.15.2), and dendritic cells by expression of CD11c+ (N418)I/A-I/E+F4/80− cells. Dead cells were excluded with live/dead stain, and the mean fluorescent intensity of each population was assessed. Data were acquired on a Cyto Flex LX (Beckman Coulter) and analyzed with FCS Express (De Novo Software). Antibody information is listed in Supplementary Table 2.

**Immunohistochemistry.** Resected tumors were fixed in 10% buffered formalin for 24 h. Tissues were processed for standard paraffin embedding with Tissue -TEK VIP6 (Sakura). Formalin-fixed, paraffin embedded tumors were cut onto slides with a thickness of 5 µm. Slides were baked for 1 hour at 60 °C, deparaffinized in xylene, and rehydrated via graded alcohols. Hematoxylin-Eosin (H&E) staining was performed using standard methods. For immunohistochemistry, the primary antibodies of CD8 (rabbit monoclonal; Abcam) and F4/80 (rabbit monoclonal; Cell Signaling) were used and fluorescence staining was done by anti-rabbit HRP polymer (Cell Signaling), followed with Tyramide signal application kit (Perkin Elmer). Cell nuclei were counterstained with 4,6-diamindino 2 phenylindole (DAPI) and slides mounted for analysis. Slides scanned with Olympus VS120, image analysis with Indica lab software. IFN-β were done by ACD RNAscope LS red detection kit.

**Siglec binding analysis.** The extracellular domain of PTGFRN (aa 26-832) with a C-terminal histidine tag was expressed in the Expi293 system (ThermoFisher) and purified by standard immobilized metal affinity chromatography methods. Fc-fusions to the indicated Siglec proteins (R&D Systems) were immobilized on Protein A biosensors (ForteBio) at 5 µg/mL and incubated with 2-fold serial

dilutions of PTGFRN starting at 100 μg/mL. The 100 μg/mL condition only is shown for Siglec 2, 4, and 8, to which no binding was detected. Equilibrium dissociation constants were calculated for Siglec 9, 10, and 14 using ForteBio Data Analysis software v. 10.0. The three phases separated by the vertical dotted lines are (i) baseline, (ii) association, and (iii) dissociation.

**Statistics and reproducibility.** Data were analyzed using Prism software (v. 8.1.0, GraphPad Software, La Jolla, CA, USA). One-way ANOVA was employed to determine statistical differences among multiple groups, and $t$-test was employed to determine differences between the two groups. $*P < 0.05$, $**P < 0.01$, $***P < 0.001$, $****P < 0.0001$. Conditions were considered significantly different for $P$ values less than 0.05. The sample size and biological replicates are indicated in the corresponding figure legend for each experiment.

**Reporting summary.** Further information on research design is available in the Nature Research Reporting Summary linked to this article.

## Data availability
Data supporting the findings of this work are available within the paper and its Supplementary files. RNA sequencing data have been deposited to GEO with accession number GSE168784. Source data are provided in Supplementary Data 1. All data are available upon reasonable request.

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

## Acknowledgements
We thank Benny Sorenson, James Thornton, and Jonathan Finn from Codiak Biosciences for scientific discussion and editing drafts of this manuscript. We thank CMC team in Codiak Biosciences for EV supply. We also thank Richard Klinghoffer from PreSage Biosciences and Eric Perouzel from InvivoGen for advices about CIVO platform and STING agonist, respectively. We thank the Oncodesign (Dijon, France) team for conducting Hepa1-6 tumor model. This work is funded by Codiak Biosciences.

## Author contributions
S.C.J., R.J.M., K.E., and S.S. designed the study. S.C.J. performed most of the in vivo experiments including tumor growth inhibition and gene expression analysis. S.C.J. and R.B. formulated exoSTING. R.J.M. performed flow cytometry. C.L.S. and N.L. performed in vitro PBMC assays. C.M. performed cytokine measurements and ELISPOT.

J.L. performed M1 and M2 differentiation. R.A.H. developed a LC-MS/MS quantification method for STING agonists. T.Z. conducted all histological assessments. K.Z. analyzed NanoString data. J.D. and M.G. conducted CIVO assay. K.K. performed in vivo study with S.C.J. N.L.R. performed in vitro EV uptake. A.V., S.E., K.X., J.S., and K.D. generated PTGFRN overexpression cells, maintained cells, and isolated EVs. K.D. also performed Western Blotting and Siglec binding. W.K.D. maintained and provided cells for in vivo studies. The data was interpreted, and the manuscript was written collectively by S.C.J., R.J.M., K.E., D.E.W., and S.S. with the support of other authors. All authors have given approval of the final version of the manuscript.

## Competing interests

All authors are employees of Codiak Biosciences, except J.D. and M.G. who are employees of Presage Biosciences.
