## [Peer Review File · Communications Biology]

Reviewers' comments:

Reviewer #1 (Remarks to the Author):

This manuscript presented a series of experimental studies using exosomes as a carrier for the delivery of cyclic dinucleotide (CDN) agonists of the Stimulator of Interferon Genes (STING). The engineered exosomes, exoSTING showed much better anti-tumor immunity than free CDNs. Both in vitro and in vivo results are very good and the exosome therapy technology is innovative.

There are, however, several major issues which need to be addressed.

- (1) The authors did not explain how the engineered exosomes were prepared. What were the donor cells? If they were cancer cells, would there be safety concerns in future clinic use?
- (2) How were the CDN agonists of STING loaded into the exosomes? What was the CDN content and what were the doses used in in vitro and in vivo experiments?
- (3) Synthetic nanocarriers such as liposomes (Reference 39) and polymersomes (Reference 40) have been used for CDN delivery with good anti-cancer activities. The authors should explain the advantages of using exosomes as the drug carrier.
- (4) Only intra-tumoral administration was used for xenograft mice in the in vivo study, which has limited translational impact. Will exoSTING function well by other delivery methods, e.g. intravenous or intraperitoneal administration, in orthotopic animal models?
- (5) In Fig. 6, the authors compared exoCDN and free CDN, and found very different effects on activation of T/NK and monocytes. Besides the activation markers CD69/CD86, did the authors measure the intracellular CDN content in each cell type (Sup Fig. 13 just shows indirect evidence by using GFP)? And what is the function of the artificial surface PTGFRN in the immune cells?

Reviewer #2 (Remarks to the Author):

In this paper the Authors describe ExoSTING, an engineered exosome that expresses prostaglandin F2 receptor negative regulator (PTGFRN) on the surface and contains Cyclic dinucleotide (CDN). ExoSTING activates the APCs in the tumor microenvironment and when used for in vivo experiments it is retained within the tumor and generates a stronger systemic anti-tumor immunity than the free CDN. Moreover, ExoSTING does not induce systemic inflammatory cytokines when used at therapeutically active doses resulting in a valid therapeutic candidate.

The Introduction is well written with recent citations to present the background and the focus of the work.

The results are logically shown.

The discussion is properly written.

Major points:

-How many different cell lines have been used to isolate exosomes? Are the data obtained from at least two different cell lines?

-More details regarding the experimental models should be added in the Results paragraph to help the readers to be focused

-Please, demonstrate the stability of the free CDN1 or CDN2

-Please, add the treatment with only exosomes without CDN, as control, for the crucial experiments

-In the "exosome isolation" paragraph in Supplementary information, please add the cell line/s that produced the exosomes

Minor points:

- "resulting in an enhanced the therapeutic window", please review the English editing

Response to Reviewers

Reviewer 1	
Comments from Reviewers	Responses
(1) The authors did not explain how the engineered exosomes were prepared. What were the donor cells? If they were cancer cells, would there be safety concerns in future clinic use?	Exosomes that were used in majority of experiments were purified from PTGFRN overexpressed HEK293 cells by using differential ultracentrifugation and density gradient. HEK293 lineage was selected as the substrate for exosome production specifically for its well-established documentation of subject safety associated with the cell line (Dumont et al. Crit. Rev. Biotechnol. 36, 1110–1122, 2016) and the capacity to conduct cell culture in chemically defined medium devoid to contaminating exosomes from animal serum. Full proteomic profile and levels of oncoprotein levels in the purified exosomes have been characterized to support the initiation of clinical trial. Details about method including engineering of cells and isolation of exosomes were described in a “Methods” and “Methods” is moved to main text according to Journal guidelines. This information is also added in a “Results” (page 4, lines 8-28) and Supplementary Fig 1.
(2) How were the CDN agonists of STING loaded into the exosomes? What was the CDN content and what were the doses used in in vitro and in vivo experiments?	CDN agonists were loaded into exosomes via passive diffusion and unloaded agonists were removed by ultracentrifugation. The amount of loaded agonists was measured by LC-MS/MS. The number of CDNs per exosome was 1189 ± 382 or 988 ± 339 for CDN1 or CDN2, respectively. The doses used in in vitro and in vivo experiments were based on the amount of CDNs and these amounts were described in the figures and manuscript. Details about loading and quantification of agonist were described in a “Methods”. This information is also added in a “Results” (page 4, lines 29-35).
(3) Synthetic nanocarriers such as liposomes (Reference 39) and polymersomes (Reference 40) have been used for CDN delivery with good anti-cancer activities. The authors should explain the advantages of using exosomes as the drug carrier.	One of the key advantages of the exosome mediated delivery is leveraging the nature selectivity mediated by complex glycoproteins on the membrane surface of exosomes and reduced toxicity and lack of immunogenicity. Nanocarriers have been used for CDN delivery with good anti-cancer activities. However, there are concerns about the cell type selectivity and toxicity associated with synthetic nanoparticles. We evaluate the uptake of DiD labelled liposomes and compared it to exosomes. We did not observe the selective uptake in APCs with liposome formulation as compared to the exosome. We evaluated the viability of macrophages after treating with liposomal CDN (according to reference

	38), exosomal CDN, and free CDN in vitro. Exosomal CDN did not reduce the viability of cells, but both liposomal CDN and free CDN reduced the viability in dose-dependent manner. This data strengthens the safety of exosomes compared to other synthetic nanoparticles. New data is added as Supplementary Figure 13 and description is added in a “Discussion” (page 13, line 37 to page 14, line 6).
(4) Only intra-tumoral administration was used for xenograft mice in the in vivo study, which has limited translational impact. Will exoSTING function well by other delivery methods, e.g. intravenous or intraperitoneal administration, in orthotopic animal models?	Majority of the IV administered exosome is taken up by the liver. To examine the activity of exoSTING by systemic administration, exoSTING was administered intravenously into mice having a mouse orthotopic hepatocarcinoma (Hepa1-6). In the results, exoSTING showed 3/8 complete response and 1/8 partial response, whereas equivalent free CDN showed no complete and partial response. This activity was correlated with biodistribution of exosomes and PD responses in the liver after IV injection. It should be noted that free CDN due to the polar nature of the compound is rapidly cleared renally and does not induce IFN-β in the liver. These data are added in new Figure 8 and related description is added in a “Results” (page 11, lines 9-31).
(5) In Fig. 6, the authors compared exoCDN and free CDN, and found very different effects on activation of T/NK and monocytes. Besides the activation markers CD69/CD86, did the authors measure the intracellular CDN content in each cell type (Sup Fig. 13 just shows indirect evidence by using GFP)? And what is the function of the artificial surface PTGFRN in the immune cells?	Direct measurement of CDN in the cytosol fraction of specific cell type would be the best answer for preferential cellular delivery by exosomes. We have developed a mass spec-based quantification assay for measuring CDN in biological samples. This method is still not sufficiently sensitive enough for measuring intra-cellular concentration of CDN (LLOQ of assay is 350 ng/mL). Ideal method will utilize a radio-labelled CDN, we have not been able to synthesize such a compound due to the challenges in the chemical synthesis. Therefore, we used the exosome associated GFP as a surrogate to understand exosome association with the different cells, although that signal is not a clear reflection of the intra-concentration mediated by endosomal escape. We believe that PTGFRN enables endosomal escape by enabling scavenger receptor and sialic acid binding receptor mediated internalization. We have identified SIGLEC family of receptors as novel binding partners to PTGFRN. Additional data is in Supplementary Fig. 12 and description is added in “Results” (page 12, lines 15-19). Although these results are suggestive of the mechanism of action of PTGFRN, a detailed evaluation of the role of scavenger receptors and SIGLEC proteins in enabling PTGFRN mediated endosomal escape is currently ongoing. In addition to PTGFRN other

	exosome surface proteins like IGSF8 and Phosphotidyl serine (PS) may also play a role in the preferential delivery of CDN to APCs.
Reviewer 2	
Comments from Reviewers	Responses
(1) How many different cell lines have been used to isolate exosomes? Are the data obtained from at least two different cell lines?	Please refer the response to reviewer 1's comment 1. Majority of exosomes that were used in the manuscript were isolated from a PTGFRN engineered HEK293 cell due to druglike properties, regulatory familiarity, and higher exosome yield. We also tested with mesenchymal stem cell (MSC)-derived exosomes. In vitro PMBC assay data with exoSTING-MSC is added in Supplementary Fig. 3d and description is added in a "Results" (page 4, lines 41-42). It should be noted that at this point we have developed scalable manufacturing for only the HEK293 derived exosomes.
(2) More details regarding the experimental models should be added in the Results paragraph to help the readers to be focused	We have now clarified these by adding more information about the experimental models throughout the manuscript and additions were highlighted with a red color. Specifically, at page 5, line 12, lines 16-17, lines 30-31, lines 33-34; page 6, line 4, lines 30-31; page 7, line 9, line 25; page 8, lines 1-3, line 34. We hope these are helpful to the readers.
(3) Please, demonstrate the stability of the free CDN1 or CDN2	Free CDN 1 and 2 were purchased from MedChem Express and Invivogen, respectively. According to the product sheet, those CDNs are stable for 6 months. exoCDN1 that was stored in -80°C for 1 year was tested in in vitro PBMC assays to examine the stability of CDN on exosomes. This exoCDN1 maintained its potency, suggesting stability of exoSTING at -80°C for up to a year. Data and description are added in new Supplementary Fig. 3c and a "Results" (page 4, lines 39-41), respectively.
(4) Please, add the treatment with only exosomes without CDN, as control, for the crucial experiments	The exosomes derived from healthy HEK293 cells, highly purified using density gradient centrifugation did not induce any immune stimulatory effects or gene-expression changes both in vitro and in vivo . We have added control experiments with empty exosomes and highlighted its immune silent activity in each of the crucial assays. Empty wild type exosomes did not activate naïve T cells, B cells and NK cells across a range of exosome concentrations (Supplementary Fig. 1e). Unloaded empty exosomes did not induce IFN- β production in the in vitro PBMC assay (Supplementary Fig. 3a). Empty exosomes in B16F10 and CT26.wt tumor models did not inhibit tumor growth (Supplementary Fig. 4e, f).

	Empty exosomes administered intra-tumorally in B16F10 model did not induce marked gene expression changes by RNA seq analysis, highlighted in Fig. 5a-c. Lastly, we further confirmed the lack of INF-β, CXCL9, and CXCL10 mRNA induction by empty exosomes with Nanostring analysis (Supplementary Fig. 6a). Overall, our highly purified empty exosomes from HEK293 cells had no intrinsic immune stimulatory activity in any of assays we've tested.
(5) In the "exosome isolation" paragraph in Supplementary information, please add the cell line/s that produced the exosomes	Exosomes were purified from PTGFRN overexpressing, wild-type, or PTGFRN knock-out HEK293 cells. This information has added in a "Methods" and "Methods" is moved to main text according to Journal guidelines.
(6) "resulting in an enhanced the therapeutic window", please review the English editing	We thank reviewer for this comment. We have checked and reviewed the English editing throughout the manuscript.

Reviewers' comments:

Reviewer #2 (Remarks to the Author):

The Authors properly revised the manuscript addressing all the reviewers' comments. The paper is definitely improved and ready for the final submission.

Reviewer #3 (Remarks to the Author):

Here the authors use extracellular vesicles loaded with cyclic dinucleotides (CDN) agonist of the STING pathway. The authors show that CDN-loaded vesicles are more potent to activate antigen presenting cell using ex- and in vivo models. Importantly they show that intra tumoral injection of CDN+-vesicles favors tumor growth inhibition, without triggering inflammatory cytokine, limiting unwanted cytotoxicity of the cells/tissues present in the tumor microenvironment.

This study has been evaluated previously by two reviewers and the authors addressed most of the concerns/comments. However, one important comment related to the mechanism of delivery of CDN associated with Extracellular vesicle could not be addressed. The authors could not measure the amount of CDN delivered within the cytosol of acceptor cells, which seems to be below the limit of detection. This is a major caveat because this prevents to formally established that CDN is indeed properly delivered in the target cells. However, this does not impact the in vivo antitumoral phenotype, the strength of the study.

In other words, the observed phenotypes are unquestionable, but the mechanism of action (cytosolic delivery via exosome/EVs) proposed by the authors remains to be formally demonstrated.

Therefore, the authors may temper their conclusions on the mechanism of action.

One possibility would be to remove most of the data that were used to tentatively demonstrate the delivery mechanism, which lack sufficient rigors and remains inconclusive. For instance, the authors might consider removing the assessment of acceptor cell positives for GFP signal emanating from GFP tagged vesicles. This only address EV uptake but not delivery. On the same line, the authors introduce at the end of the study the putative role of their proprietary EV marker named PTGFRN. Although it is highly possible that glycan/protein interaction might favor EV uptake and perhaps delivery, as previously suggested, the experiments proposed here do not constitute a direct proof and somehow perturb the main message of the paper.

To conclude on the mechanistical aspect of the study, the authors compared ex-vivo and in vivo the potency of EXOCDN1 vs. CDN1 alone and vs. CDN1 + free exosome (figure 1 C). Since loading is performed via passive diffusion over 24h, why then the condition "free exo +CDN1" (which are incubated with target PBMC cells for 24 hours) does not at least partially increases CDN1 efficiency ? During the incubation with the target cells, one would expect that at least a portion of the free exosomes can capture a portion of CDN1 and enhance their capacity to activate the target cells. Could the authors explain why this is not the case?

On the same line if the cell response to DPN1 is truly enhance by exosome as a carrier, one would expect that free exosome may compete with CDN1-loaded exosomes and decrease the efficiency of the latter. Therefore, the authors must experimentally test at least ex vivo (in figure 1C) if free exosome compete with CDN 1 loaded exosome and explain the results, especially if free exosome do not compete.

Others comments:

-The authors might consider using the term EVs instead of exosome since they did not formally established MVB- as the source of the vesicles. This will surely avoid endless debate about the nomenclature that could be detrimental to the future citation of the paper. At least the authors

should mention that they use exosome as a generic term.

-in supplementary figure 1, in which the authors characterized the EVs , the authors should show a negative control for EV (i.e ER or Golgi proteins) and systematically show all the marker in cell lysate vs. EVs

-It has been shown that EV loaded with cGAs were not efficient to activate the STING pathway whereas EV loaded with cGAS and harboring VSV-G were efficient (Gentili et al, science, 2015). This seems contradictory with the findings reported here and should be discussed.

-Since the authors do not establish that CDN are inside the vesicles, they should clearly mentioned that CDNs are associated with EVs and could be inside and outside of the vesicles.

Response to Reviewers

Reviewer 3	
Comments from Reviewers	Responses
(1) To conclude on the mechanistical aspect of the study, the authors compared ex-vivo and in vivo the potency of EXOCDN1 vs. CDN1 alone and vs. CDN1 + free exosome (figure 1 C). Since loading is performed via passive diffusion over 24h, why then the condition “free exo +CDN1” (which are incubated with target PBMC cells for 24 hours) does not at least partially increases CDN1 efficiency? During the incubation with the target cells, one would expect that at least a portion of the free exosomes can capture a portion of CDN1 and enhance their capacity to activate the target cells. Could the authors explain why this is not the case?	Our studies show that loading of CDN is temperature, time and CDN concentration dependent. We have established that a high concentration of CDN is required for loading the EVs effectively ($\geq 1\text{mM}$). The maximal concentration of the CDN1 used in the free EV+CDN1 group is $100\ \mu\text{M}$ which is about 10-fold lower than the concentration required for loading the EVs. Therefore, we don't expect any EV mediated delivery of CDN1 when just co-incubating free EVs with CDN1 at that low CDN concentrations.
(2) On the same line if the cell response to DPN1 is truly enhance by exosome as a carrier, one would expect that free exosome may compete with CDN1-loaded exosomes and decrease the efficiency of the latter. Therefore, the authors must experimentally test at least ex vivo (in figure 1C) if free exosome compete with CDN 1 loaded exosome and explain the results, especially if free exosome do not compete.	We thank the reviewer for this suggestion. We have the competition of exoSTING with unloaded EVs. Indeed, we observed a dose-dependent inhibition of IFN-β production in the competition assay with unloaded EVs, suggesting that EVs loaded with CDN are required for exoSTING potency. These results confirm EV mediated delivery of CDN. This information is added in the “Results” (page 5, lines 7-11) and Supplementary Fig 3f.
(3) The authors might consider using the term EVs instead of exosome since they did not formally established MVB- as the source of the vesicles. This will surely avoid endless debate about the nomenclature that could be detrimental to the future citation of the paper. At least the authors should mention that they use exosome as a generic term.	We changed the term of “exosomes” to “EVs” throughout the manuscript.
(4) in supplementary figure 1, in which the authors characterized the EVs, the authors should show a negative control for EV (i.e ER or Golgi proteins) and systematically show all the marker in cell lysate vs. EVs	Now, we completed the Western Blot again with cell lysate control and negative control of EVs (ER protein, calnexin). This information is added in a “Results” (page 4, lines 17-18) and Supplementary Figure 1d. In addition, unprocessed whole images are presented in Supplementary Figure 15.
(5) It has been shown that EV loaded with cGAS were not efficient to activate the STING pathway whereas EV loaded with cGAS and harboring VSV-G were efficient (Gentili et al, science, 2015). This seems contradictory with the findings reported here and should be discussed.	Gentili et al showed that cGAS can be delivered via EVs derived from infected cells. The activity EVs were weak and could be enhanced by VSV-G. This may be mainly due to the low loading efficiency of cGAS in EVs as they haven't quantified the amount of cGAS. In addition, fusogenic peptides like VSV-G may be required for endosomal escape of a large

	protein like cGAS (62kD) for efficient cytosolic delivery. On the other hand, in this study we loaded EVs with a small molecule CDN STING agonist (~700 Da), which does not require a fusogenic peptide for endosomal escape. Therefore, we are able to induce the sufficient activation of STING pathway without VSV-G. Previous reports of EV mediated delivery of microRNAs and ASOs without the need for fusogenic peptide has been documented (Neurobiol Dis. 2021, 148:105218; Eur J Pharm Biopharm. 2021, 158:198-210; Int J Nanomedicine. 2018, 13: 7727–7747; Nat Rev Mol Cell Biol. 2020, 21:585-606). This is now added to the discussed in page 14, lines 1-4.
(6) Since the authors do not establish that CDN are inside the vesicles, they should clearly mentioned that CDNs are associated with EVs and could be inside and outside of the vesicles.	We agree with the reviewer comment. We added a new sentence in the “Results” section, page 4, lines 33-34, explaining that CDNs can be associated with both inside and/or outside of EVs.

REVIEWERS' COMMENTS:

Reviewer #3 (Remarks to the Author):

Congratulations.